# Set Prediction in the Latent Space

**Konpat Preechakul**[1]            **Chawan Piansaddhayanon**[1]            **Burin Naowarat**[1]

**Tirasan Khandhawit**[2]            **Sira Sriswasdi**[3]            **Ekapol Chuangsuwanich**[1,3]

[1]Department of Computer Engineering, Chulalongkorn University
[2]Department of Mathematics, Faculty of Sciences, Mahidol University
[3]Computational Molecular Biology Group, Faculty of Medicine, Chulalongkorn University
`konpatp@gmail.com`, `{6372025021, 6270145221}@student.chula.ac.th`
`tirasan.kha@mahidol.ac.th`, `{sira.sr, ekapol.c}@chula.ac.th`

## Abstract

Set prediction tasks require the matching between predicted set and ground truth set in order to propagate the gradient signal. Recent works have performed this matching in the original feature space thus requiring predefined distance functions. We propose a method for learning the distance function by performing the matching in the latent space learned from encoding networks. This method enables the use of teacher forcing which was not possible previously since matching in the feature space must be computed after the entire output sequence is generated. Nonetheless, a naive implementation of latent set prediction might not converge due to permutation instability. To address this problem, we provide sufficient conditions for permutation stability which begets an algorithm to improve the overall model convergence. Experiments on several set prediction tasks, including image captioning and object detection, demonstrate the effectiveness of our method. Code is available at `https://github.com/phizaz/latent-set-prediction`.

## 1   Introduction

Set prediction is a task where a model predicts multiple elements whose ordering is not relevant for correctness. This task is central to many real-world problems such as object detection, image captioning, and multi-speaker speech recognition. Object detection requires predicting a set of bounding boxes without any specific ordering. Describing objects within an image is a kind of image captioning yet perfectly suitable for set prediction. Multi-speaker speech recognition is also well suited for set prediction since the order of transcripts is irrelevant. Though these tasks can naturally be modeled as set prediction, traditional deep learning is not inherently suitable for these tasks.

Multi-layer perceptrons and convolution networks with traditional loss functions impose a specific ordering on the prediction heads which hinders set prediction. A reasonable set prediction pipeline requires the model's prediction heads to be more flexible. Each head does not have a predefined target, yet relies on its peers to determine what is best to predict to complete the target set. Recent works [1, 2] emphasized using a Transformer model [3], which is permutation-invariant, coupled with a permutation-invariant loss function as the main ingredients. Any traditional loss function can be made permutation-invariant by solving for a *minimum* bijective matching between predicted set and ground truth set via the Hungarian algorithm under a certain **distance metric**. After the assignment, the loss function is calculated between the assigned pairs, and backpropagation is performed accordingly. This scheme is known as Permutation Invariant Training (PIT).

35th Conference on Neural Information Processing Systems (NeurIPS 2021).

A distance metric used by the assignment must agree with the loss function in a way that the assignment is kept after an optimization step on the loss function. A distance metric that fails this criterion may switch pairings hindering the convergence. Hence, a distance metric is crucial to the convergence property of the set optimization scheme. One may argue to use the loss function itself as a distance metric. However, not all loss functions have meaningful scalar values. For example, the vanilla GAN's loss [4] is not insightful in terms of progress or distance. Combining loss functions from different domains also complicates the matter because they are not easily comparable in their scalar forms. In object detection, both L1 error loss and cross entropy loss are used to learn bounding box prediction [2], but it is unclear how to define a proper distance metric from such a combination. Either hand-tuned coefficients or different surrogate distance metrics may be needed to form an effective distance metric. This begets the problem of selecting a proper distance metric for PIT. A set prediction scheme that does not require a hand-tuned distance function is appreciable.

Another hardship related to PIT is when applying set prediction on sequence domains that require *teacher forcing* to train. Auto-regressive with teacher forcing is often used for sequence prediction such as speech recognition [5, 6] and machine translation [7, 3]. However, teacher forcing requires a groundtruth assignment before it can begin prediction. PIT also relies on the teacher forcing prediction to do minimum assignment, resulting in a chicken and egg problem. If the set cardinality is small enough, it is possible to exhaustively teacher force with respect to all possible ground truths requiring $O(N^2)$ forward passes through the model, and keep only those with the minimum assignment distances for optimization.

What if the Hungarian assignment is done in a latent space instead? Since the latent space is learned, the choice of any specific distance metric is alleviated – even a simple Euclidean distance is reasonable. Since the latent space is prior to the sequence prediction, the prediction process knows exactly what its ground truth is which allows for efficient, $O(N)$, teacher forcing. This paper presents **latent set prediction** (LSP) which enables the assignment in the latent space with Euclidean distance metric. At the same time, it provides a convergence guarantee of the loss function by reducing the effect of permutation switches that can be problematic when performing matching in the latent space. Our contributions are as follows:

1. We propose a framework for deep set prediction that alleviates the need for hand-crafted distance metrics.

2. This framework is efficient for the set of sequence predictions with teacher forcing requiring only $O(N)$ predictions, an improvement from the usual exhaustive $O(N^2)$.

3. We provide a convergence proof of set prediction under this framework.

## 2  Related works

**Set prediction.** There are mainly two families of set prediction: distribution matching and minimum assignment. The distribution matching approaches learn $P(Y|x)$ where $Y$ is a set and $x$ is an input. DeepSetNet and variants [8–10] proposed a likelihood function for set prediction. An energy function learned via adversarial samples was also proposed [11]. On the other hand, the minimum assignment approaches rely on solving assignment problems. The loss function is calculated between the assigned pairs afterward. Either Hungarian assignment (bijection) or Chamfer assignment is usually used depending on tasks. Zhang et al. [12] proposed to *mold* a primitive set into a target set via gradient signals from a set encoder. Kosiorek et al. [1] proposed a Transformer for set prediction. A similar kind of design was also used in end-to-end object detection [2]. Besides the two assignments, a stable marriage was proposed for set autoencoding pretraining [13].

**Image captioning** is not usually related to set prediction. This is true for *impression* captions such as MS-COCO [14] which only describe the most salient objects. A different kind of captioning is *descriptive* which describes individual objects in a scene and their interactions. A prominent example is Visual Genome [15]. For the same reason, a chest radiology report is also descriptive [16]. Since descriptive captions have no specific ordering, this task is actually a set prediction where the elements are captions themselves. To the best of our knowledge, there is no practical approach for set of text predictions that involves *teacher forcing*.

**Object detection** is formulated as a set prediction task where each bounding box is a set member. Most object detection algorithms impose ordering by dividing the image into several grids. Each cell

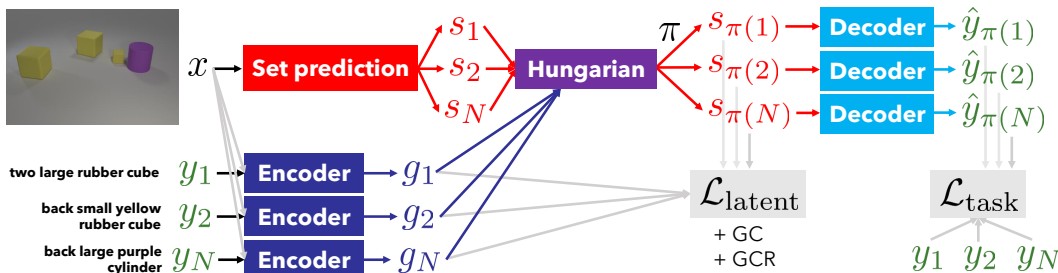

Figure 1: Latent set prediction (LSP) framework where $x$ is an image and $y$'s are sentences (it can be applied to any $x$ and $y$). The Hungarian algorithm is used to find the minimum assignment between $s$'s (predictions) and $g$'s (encoded $y$'s). This allows efficient *teacher forcing* at decoder $\mathcal{D}$ which is not possible previously. The latent loss $\mathcal{L}_{\text{latent}}$ is applied to minimize the distance between the $s$-$g$ pairs. The task loss $\mathcal{L}_{\text{task}}$ is applied as usual on the prediction. Only set prediction model and decoder are required during inference.

is responsible for predicting bounding boxes whose centers reside in it. Anchors are introduced to allow a single cell to support multiple objects [17, 18]. As a result, hand-crafted components are needed for these methods to function efficiently. Though many works are focusing on removing the use of anchors [19, 20], the dense grid prediction still remains. Later, DETR [2] directly applies set prediction on bounding boxes whose process matches predicted boxes to ground truth boxes. To achieve a satisfactory result, the matching cost has to be manually designed.

## 3 Latent Set Prediction (LSP)

A common set prediction pipeline has three components: a set prediction model, ground truths, and an assignment mechanism. Our method is focused on the case of Hungarian assignment. Traditionally, the assignment mechanism matches the ground truths with the model predictions in an *output space* $Y$. Here, the pairing happens in a **latent space** $\mathbb{R}^C$.

### 3.1 Notations

We assume a **set prediction model** $\mathcal{F} : X \rightarrow \mathbb{R}^{N \times C}$ where $X$ is the input space and $N$ is the cardinality of the set. Effectively, $\mathcal{F}$ outputs $N$ vectors in a latent space $\mathbb{R}^C$. The model $\mathcal{F}$ is also responsible for set cardinality prediction. Each latent vector is passed through a **decoder** $\mathcal{D} : \mathbb{R}^C \rightarrow Y$ where $Y$ is any output space. Note that the **decoder** may also accept the input $x \in X$ wherever the input is required for better prediction. To pair in the latent space, we utilizes another component called **encoder** $\mathcal{E} : Y \times X \rightarrow \mathbb{R}^C$ where $y \in Y$ is an output and $x \in X$ is its corresponding input. The encoder maps elements of the output space as **guiding vectors** $g \in \mathbb{R}^C$ in the latent space to facilitate the assignment. Given a set of latent vectors $\{s_1, s_2, \ldots, s_N\}$ and guiding vectors $\{g_1, g_2, \ldots, g_N\}$, the minimum assignment $\pi$ is

$$\pi = \operatorname*{argmin}_{\pi' \in \mathcal{P}} \sum_{i}^{N} \|s_{\pi'(i)} - g_i\|_2 \qquad (1)$$

where $\mathcal{P}$ is the set of all permutations of $N$ letters.

A **switch** is said to occur when $s_{\pi(i)}$ changes after a gradient update as illustrated in Figure 2.

### 3.2 Method

Latent set prediction (LSP) begins by feeding an input $x$ into the **set prediction model** $\mathcal{F}$ resulting in a set of **latent vectors** $s$'s. $s$'s do not have designated targets until the corresponding **guiding**

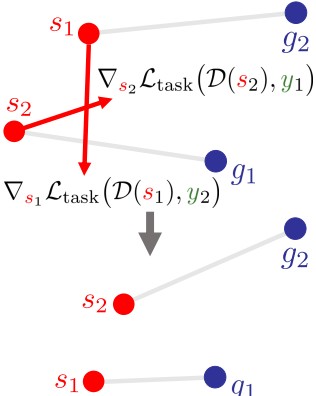

Figure 2: $s$'s move toward $y$'s designated by $g$'s. At the same time, $s$'s may move away from their $g$'s resulting in a **switch**.

vector $g$'s are retrieved. To get $g$, each **ground truth** $y_i$ is mapped via the **encoder** resulting in $g_i$. The assignment algorithm is performed between $s$'s and $g$'s resulting in the **minimum bijective matching** $\pi$. This pairs up each latent vector $s_{\pi(i)}$ to the guiding vector $g_i$ and the associated ground truth $y_i$. Knowing its target, $s$ goes through the **decoder** $\mathcal{D}$ resulting in $\hat{y}$. The **task loss function** $\mathcal{L}_{\text{task}}$ is calculated accordingly between $\hat{y}_{\pi(i)}$ and $y_i$, and then the optimization is performed. Note that $s$ receives no training signal from $g$; $g$ only gives $s$ its goal. After an optimization step, as $s_{\pi(i)}$ moves along the task gradient toward better prediction of $y_i$, it may move *away* from $g_i$ and approach another guiding vector $g_j$. This can cause a **switch** as demonstrated in Figure 2. Our method incorporates several techniques that encourage stable pairing of $s$'s and $g$'s over time, which turns out to be sufficient for the convergence of $\mathcal{L}_{\text{task}}$. A pictorial description of the LSP framework is depicted in Figure 1.

The proof of convergence of $\mathcal{L}_{\text{task}}$ (Section 4) indicates that the convergence hinges on the ever smaller gaps between $s$'s and $g$'s under $\pi$. In fact, $\mathcal{L}_{\text{task}}$ is bounded from above by a function of $\sum_i^N \left\| s_{\pi(i)} - g_i \right\|_2$. Hence, not only that $g$'s give $s$'s their goals, $g$'s must also follow wherever $s$'s go. By closing the gaps, it is less likely for a switch to happen. Should a switch happen, it would only be between a short distance which is not as harmful to $\mathcal{L}_{\text{task}}$. This does not imply that we need to avert switches at all costs. We can simply assign $g_i$ to $s_i$ for all $i$ to guarantee no switches. However, it is ordered prediction, not set prediction. In a sense, switches should be welcomed as a sign of learning a *natural* ordering as long as in the long run the gaps are still closing.

Therefore, we propose two mechanisms to make sure that the gaps between $s$'s and $g$'s are smaller over time. First, we enforce an **asymmetric latent loss** to bring $s$'s and $g$'s together:

$$\mathcal{L}_{\text{latent}}^{s \to g} = \sum_i \frac{1}{2} \| s_{\pi(i)} - [g_i] \|_2^2 \quad \mathcal{L}_{\text{latent}}^{g \to s} = \sum_i \frac{1}{2} \| [s_{\pi(i)}] - g_i \|_2^2 \tag{2}$$

$$\mathcal{L}_{\text{latent}} = \beta \mathcal{L}_{\text{latent}}^{s \to g} + \gamma \mathcal{L}_{\text{latent}}^{g \to s} \qquad \mathcal{L}_{\text{total}} = \mathcal{L}_{\text{latent}} + \mathcal{L}_{\text{task}}$$

where $[\cdot]$ is stop gradient, and $\beta, \gamma$ control the loss strengths. Ideally, we want to set $\beta = 0$ since $s$'s should only follow the training signals from $\mathcal{L}_{\text{task}}$. However, we found $\beta = 0.1$ to be useful in practice providing a bit of help for $g$'s to meet $s_\pi$'s. We set $\gamma = 1$ as the default value and found it to work well across experiments.

However, the latent loss alone is not enough to guarantee convergence. This is because the latent loss cannot anticipate the movement of $s$'s. Even for a pair of infinitesimally close $s$ and $g$, any sizeable $\nabla_{s_\pi} \mathcal{L}_{\text{task}}$ can break apart the two. The second part which completes the convergence proof is **gradient cloning** (GC) which copies the task gradient $\nabla_{s_\pi} \mathcal{L}_{\text{task}}$ from $s_\pi$'s to their respective $g$'s. Theoretically, the distance between $s$ and $g$ is strictly decreasing which satisfies the requirement for convergence.

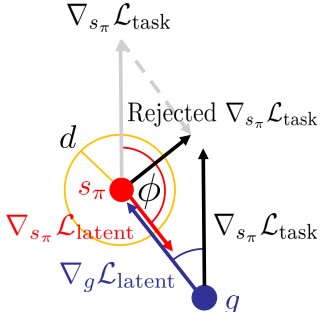

In practice, the models that predict $s$'s and $g$'s may not be equally capable as one may go faster than the other. To allow for this discrepancy, we propose a stronger version of GC namely **gradient cloning with rejection** (GCR). With GCR, the **leader** of each pair of $s_\pi$ and $g$ is *slowed* down when $\| \nabla_{s_\pi} \mathcal{L}_{\text{latent}} - \nabla_g \mathcal{L}_{\text{latent}} \|_2$ is larger than $d \| \nabla \mathcal{L}_{\text{task}} \|_2$. The constant $d$ (default $d = 10^{-3}$) is indicating whether $s$ and $g$ are sufficiently far apart (relative to $\| \nabla \mathcal{L}_{\text{task}} \|_2$) requiring a slower leader. The leader is slowed down by rejecting its $\nabla \mathcal{L}_{\text{task}}$ along the span of $\nabla \mathcal{L}_{\text{latent}}$. The one with an *obtuse* angle between its task and latent gradients is considered a leader: $\langle \nabla_{s_\pi} \mathcal{L}_{\text{task}}, \nabla_{s_\pi} \mathcal{L}_{\text{latent}} \rangle < 0$ (in case of $s$) or $\langle \nabla_{s_\pi} \mathcal{L}_{\text{task}}, \nabla_g \mathcal{L}_{\text{latent}} \rangle < 0$ (in case of $g$). $d$ serves as a parame-

Figure 3: GC with rejection. $s$ is the leader in this case since its $\nabla \mathcal{L}_{\text{task}}$ and $\nabla \mathcal{L}_{\text{latent}}$ form an obtuse angle. Its $\nabla \mathcal{L}_{\text{task}}$ is rejected along the span of its $\nabla \mathcal{L}_{\text{latent}}$ when its bidirectional latent gradient's length exceeds $d \| \mathcal{L}_{\text{task}} \|_2$.

ter for choosing between GC ($d = \infty$) and GCR with always rejection ($d = 0$). A smaller $d$ puts a stronger tendency for converging $s$ and $g$ at the cost of slower learning of $s$. The idea is depicted in Figure 3 and described in Algorithm 2.

We summarize LSP in Algorithm 1 which can be implemented efficiently with modern deep learning frameworks.

---

**Algorithm 1** Single training step of Latent Set Prediction (LSP)

---

Given $\mathbf{x} \in X, \mathbf{Y} \in Y^N, d \in \mathbb{R}$
$\mathbf{S} \leftarrow \mathcal{F}(\mathbf{x})$             $\triangleright \mathbf{S} \in \mathbb{R}^{N \times C}$, latent set element prediction

---

(Inference only)
$\hat{\mathbf{Y}} \leftarrow \mathcal{D}(\mathbf{S})$             $\triangleright \hat{\mathbf{Y}} \in Y^N$, prediction on output space

---

(Training only)
$\mathbf{G} \leftarrow \mathcal{E}(\mathbf{Y}, \text{repeat}(\mathbf{x}))$          $\triangleright \mathbf{G} \in \mathbb{R}^{N \times C}$, ground truth encoding
$\pi \leftarrow \text{Hungarian}(\mathbf{G}, \mathbf{S})$          $\triangleright$ Equation 1
$\hat{\mathbf{Y}}_\pi \leftarrow \mathcal{D}(\mathbf{S}_\pi)$          $\triangleright \hat{\mathbf{Y}}_\pi \in Y^N$, prediction on output space
$\mathcal{L}_{\text{latent}} \leftarrow \mathcal{L}_{\text{latent}}(\mathbf{S}_\pi, \mathbf{G})$          $\triangleright$ Equation 2
$\mathcal{L}_{\text{task}} \leftarrow \mathcal{L}_{\text{task}}(\hat{\mathbf{Y}}_\pi, \mathbf{Y})$
$\nabla_{\mathbf{S}_\pi} \mathcal{L}_{\text{task}}, \nabla_{\mathbf{S}_\pi} \mathcal{L}_{\text{latent}}, \nabla_{\mathbf{G}} \mathcal{L}_{\text{latent}} \leftarrow \text{Backprop}(\mathcal{L}_{\text{task}} + \mathcal{L}_{\text{latent}})$
$\nabla_{\mathbf{S}_\pi} \leftarrow \text{GCR}(\nabla_{\mathbf{S}_\pi} \mathcal{L}_{\text{task}}, \nabla_{\mathbf{S}_\pi} \mathcal{L}_{\text{latent}} - \nabla_{\mathbf{G}} \mathcal{L}_{\text{latent}}, d)$      $\triangleright$ Algorithm 2
$\nabla_{\mathbf{G}} \leftarrow \text{GCR}(\nabla_{\mathbf{S}_\pi} \mathcal{L}_{\text{task}}, \nabla_{\mathbf{G}} \mathcal{L}_{\text{latent}} - \nabla_{\mathbf{S}_\pi} \mathcal{L}_{\text{latent}}, d)$      $\triangleright$ Algorithm 2
Continue backpropagation to $\mathcal{F}, \mathcal{D}, \mathcal{E}$'s parameters

---

---

**Algorithm 2** Gradient Cloning with Rejection (GCR)

---

Given $\nabla \mathcal{L}_{\text{task}}, \nabla \mathcal{L}_{\text{latent}}$, and $d \in \mathbb{R}$
$\text{obtuse} \leftarrow \langle \nabla \mathcal{L}_{\text{task}}, \nabla \mathcal{L}_{\text{latent}} \rangle < 0$      $\triangleright \text{obtuse} \in [0,1]^N$, obtuse angles indicate leading positions
$\text{far} \leftarrow \|\nabla \mathcal{L}_{\text{latent}}\|_2 > d \cdot \|\nabla \mathcal{L}_{\text{task}}\|_2$      $\triangleright \text{far} \in [0,1]^N$, large latent gradients indicate large distances
$\hat{\nabla} \mathcal{L}_{\text{latent}} \leftarrow \frac{\mathcal{L}_{\text{latent}}}{\|\mathcal{L}_{\text{latent}}\|_2}$
$\nabla \mathcal{L}_{\text{task}} \leftarrow \nabla \mathcal{L}_{\text{task}} - (\text{obtuse} \cdot \text{far}) \cdot \hat{\nabla} \mathcal{L}_{\text{latent}} \cdot \langle \nabla \mathcal{L}_{\text{task}}, \hat{\nabla} \mathcal{L}_{\text{latent}} \rangle$      $\triangleright$ Gradient rejection
Return $\nabla \mathcal{L}_{\text{task}} + \nabla \mathcal{L}_{\text{latent}}$

---

## 4 Convergence analysis

In this section, we show that **gradient cloning** (GC) technique together with a special case of **asymmetric latent loss**, $\beta = 0$ in (2), is sufficient for the convergence of LSP to a local minimum under the following mild assumptions:

1. Each latent vector $s_i$ and each guiding vector $g_i$ is updated according to the gradients exactly as expressed in the training dynamics defined in Section 4.1.

2. $\mathcal{L}_{\text{task}}$ is **L-smooth** and satisfies the **Polyak-Lojasiewicz** condition. This is typically assumed to prove the convergence of the gradient descent algorithm [21].

Under the standard gradient descent setup, the convergence of LSP is complicated by the fact that each **switch** can increase the task loss as $s_{\pi(i)}$ changes its target from $y_i$ to a new $y_j$. Our **gradient cloning** and **asymmetric latent loss** techniques ensure that even though **switch** can keep occurring throughout the model training process, its impact on task loss will decay exponentially.

Detailed proofs are provided as an Appendix for interested readers. It should be noted that similar results can be obtained for **gradient cloning with rejection** (GCR) and general cases of **asymmetric latent loss** with $\beta > 0$.

### 4.1 LSP training dynamics with gradient cloning

We begin with the explicit notations for Algorithm 1. The training at each time point $t + 1$ consists of two steps. First, the assignment $\pi$ is updated according to the values of $s_{\pi(i)}^{(t)}$'s and $g_i^{(t)}$'s from the previous time step as defined in (1). Then, the values of $s_{\pi(i)}$'s and $g_i$'s are updated based on the gradients from $\mathcal{L}_{\text{task}}$ and $\mathcal{L}_{\text{latent}}$ with **step size** $\eta$ and **latent loss strength** $\gamma$ as defined in (2).

$$s_{\pi^{(t+1)}(i)}^{(t+1)} = s_{\pi^{(t+1)}(i)}^{(t)} - \eta \nabla_s l_{\text{task}}(s_{\pi^{(t+1)}(i)}^{(t)}, y_i)$$

$$g_i^{(t+1)} = g_i^{(t)} - \eta \left( \nabla_s l_{\text{task}}(s_{\pi^{(t+1)}(i)}^{(t)}, y_i) + \gamma \nabla_g l_{\text{latent}}(s_{\pi^{(t+1)}(i)}^{(t)}, g_i^{(t)}) \right)$$

where $l_{\text{task}}(\cdot, y_i)$ subsumes the prediction head $g(\cdot)$. The lower case notations $l_{\text{task}}$ and $l_{\text{latent}}$ correspond to per-data-point loss functions.

A direct implication of gradient cloning, which attracts $s$ and $g$ together, is that the total distance $\sum_i \|s_{\pi^{(t)}(i)}^{(t)} - g_i^{(t)}\|_2$ decays exponentially. This consequently ensures that when a **switch** occurs at time $t$, the distance between involved latent vectors $\|s_{\pi^{(t+1)}(i)}^{(t)} - s_{\pi^{(t)}(i)}^{(t)}\|_2$ also decay exponentially. Hence, the impact of **switch** on $\mathcal{L}_{\text{task}}$ decreases rapidly over the course of model training.

## 4.2 Impact of a switch on task loss

The impact of a **switch** on task loss can be illustrated mathematically using the **L-smooth** condition

$$l_{\text{task}}(s_{\pi^{(t+1)}(i)}^{(t+1)}, y_i) - l_{\text{task}}(s_{\pi^{(t)}(i)}^{(t)}, y_i) \leq \frac{L}{2} d_t^2 + d_t \|\nabla_s l_{\text{task}}(s_{\pi^{(t)}(i)}^{(t)}, y_i)\|_2 - c \|\nabla_s l_{\text{task}}(s_{\pi^{(t+1)}(i)}^{(t)}, y_i)\|_2^2,$$

where $c > 0$ and $d_t = \|s_{\pi^{(t+1)}(i)}^{(t)} - s_{\pi^{(t)}(i)}^{(t)}\|_2$ is the distance between latent vectors involved in a **switch**. It should be noted that the negative gradient term appears in a typical proof of convergence for gradient descent while the terms with $d_t$ are introduced by the assignment step (1). In the absence of a switch, task loss always decreases. However, if $d_t$ is large, the first two terms can dominate.

## 4.3 Convergence of LSP

By viewing the right-hand side of the inequality above as a quadratic function of $d_t$, we can see that if the magnitude of task gradients are larger than some factor of $d_t$, then the right-hand side must be negative. This implies that the task loss decreases. On the other hand, if the task gradients are small, the **Polyak-Lojasiewicz** condition

$$\frac{1}{2} \|\nabla_s l_{\text{task}}(x, y)\|^2 \geq \mu \cdot (l_{\text{task}}(x, y) - l_{\text{task}}(x^*, y)), \text{ where } x^* \text{ is a global minimum} \qquad (3)$$

then implies that our optimization is already near a minimum. Thus, we put the two cases together to obtain the following key result.

**Theorem 4.1.** *If $l_{task}(\cdot, y)$ is **L-smooth** and satisfies the **Polyak-Lojasiewicz** condition, then*

$$l_{task}(s_{\pi^{(t+1)}(i)}^{(t+1)}, y_i) - l_{task}(s_i^*, y_i) \leq \begin{cases} C\,\alpha^{2t}, & \text{if } \|\nabla_s l_{task}(s_{\pi^{(t)}(i)}^{(t)}, y_i)\| \leq \frac{3d_t}{\eta\left(1 - \frac{L\eta}{2}\right)} \\ \delta\left(l_{task}(s_{\pi^{(t)}(i)}^{(t)}, y_i) - l_{task}(s_i^*, y_i)\right), & \text{otherwise} \end{cases}$$

*where appropriate choices of **step size** $\eta$ will ensure that all constants are positive and $\alpha$, $\delta < 1$.*

This implies that the difference between the current task loss and the global minimum is bounded above by the maximum of two sequences of positive real numbers, both converging to zero with linear rates. Therefore, the task loss of LSP converges to a local minimum with a linear rate.

# 5 Experiments

We first demonstrate that GC and GCR help reduce switches in a synthetic dataset. Without our methods, the models might not converge. Then, we apply LSP on common set prediction tasks such as object detection and point cloud prediction (see Appendix B). One unique ability that LSP enables is allowing teacher forcing in set of texts prediction scenarios which we demonstrate on a CLEVR object description task and on a challenging MIMIC chest x-ray report generation task. Without LSP, these kind of tasks were not possible to perform set prediction in due to the computational cost.

## 5.1 Synthetic dataset

This experiment aims to demonstrate the convergence properties of LSP variants (without GC, with GC, and with GCR ($d = 0$)). Three sets of $N$ random points from standard normal distribution were generated. The three sets represent $s$'s, $g$'s, and $y$'s, all in $\mathbb{R}^{\dim}$ space. $\mathcal{L}_{\text{task}}(s_\pi, y) = \|s_\pi - y\|_2^2$, i.e. no prediction head. The loss is defined as $\mathcal{L}_{\text{task}}(s_\pi, y) + \alpha \mathcal{L}_{\text{latent}}(s_\pi, g)$ where $\alpha$ serves as a relative

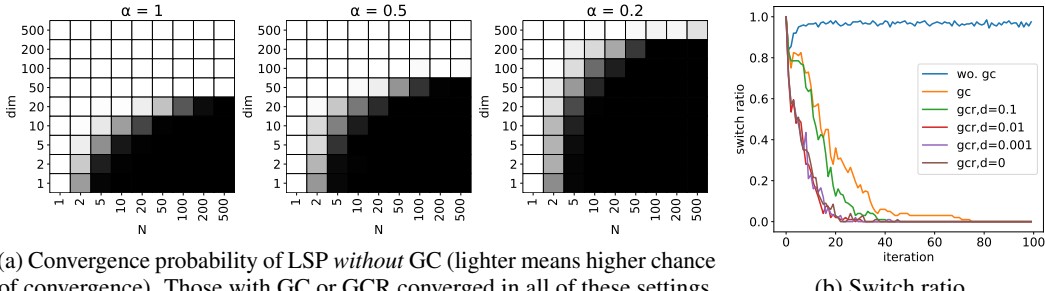

(a) Convergence probability of LSP *without* GC (lighter means higher chance of convergence). Those with GC or GCR converged in all of these settings.

(b) Switch ratio

Figure 4: Comparing LSP with/without GC and GCR on the synthetic dataset.

strength between $\mathcal{L}_{\text{task}}$ and $\mathcal{L}_{\text{latent}}$. We altered $\alpha$ to better demonstrate their behaviors in practice since both losses may be of different magnitudes. No neural networks were used in this experiment.

**Convergence probability** is how likely a trial will converge (from 100 trials). A trial is considered converged if after 300 iterations[1] $\mathcal{L}_{\text{task}} < 0.01$. A robust algorithm should always converge. Figure 4a demonstrates LSP *without* GC on different $N$'s and $\dim$'s and $\alpha$'s. The results confirmed that without GC the training may not converge while those with GC converged robustly in all of these settings. Positive factors for convergence are: smaller $N$, larger $\dim$, larger $\alpha$. In other words, keeping $s$ closer to $g$ than the other $s$'s. Larger $N$ and smaller $\dim$ reduce average spaces between points, and smaller $\alpha$ leaves a larger gap between a leading $s$ and a trailing $g$ weakening the bond.

**Switch ratio** is the fraction of $s$'s that were matched to different $y$'s after an update. Decreasing switch ratio as the training progresses is a good sign for convergence. We experimented with $N = 200, \dim = 2, \alpha = 0.5$ and penalized the gradient towards $g$'s to be 0.5 times smaller than those of $s$'s. This setting demonstrates a suboptimal encoder that cannot easily follow $s$'s. Figure 4b that, with a suboptimal encoder, GCR ($d = 0$) reduces the switch ratio the fastest due to its ability to slow down the faster $s$'s the most.

## 5.2 Object detection

We will demonstrate that LSP is applicable to common set prediction tasks such as object detection and point cloud prediction (see Appendix B). The goal is to show that LSP achieves a competitive performance compared to a manually designed assignment cost. We compared LSP against **DETR** [2], which is a reasonably strong baseline, that can be adapted to work with LSP with minimal changes. We used our modified MNIST dataset [22] in this experiment. The dataset contains 5,000 training and 1,000 test images. Each data point contains multiple randomly placed digits from the MNIST dataset. To increase the difficulty of the dataset, each digit in the image was augmented by using a random photometric distortion, morphological transformation, and random resizing. We reported test AP of the last training epoch. $\text{AP}_{\text{L}}$ was not reported because there is no large object in our dataset. See Appendix Figure 6 for example images in our dataset.

Table 1 shows that **LSP** achieved a competitive result compared to a DETR baseline. Particularly, LSP with GCR outperformed the baseline when a small value of $d$ was used. The gain primarily came from an increase in predicted bounding box quality shown by +3.6 $\text{AP}_{75}$ over the baseline. The result also suggested that **LSP** led to a more robust matching for different object sizes as the matching cost is learnt. This is contrary to a manually designed fixed

| Method | AP | $\text{AP}_{50}$ | $\text{AP}_{75}$ | $\text{AP}_{\text{S}}$ | $\text{AP}_{\text{M}}$ |
|---|---|---|---|---|---|
| DETR [2] | 43.5 | 71.0 | 49.1 | 39.8 | 64.5 |
| LSP (GC) | 31.1 | 62.3 | 26.8 | 28.3 | 48.9 |
| LSP (GCR, $d = 10^{-3}$) | 45.0 | 71.2 | 51.4 | 41.6 | 64.7 |
| LSP (GCR, $d = 10^{-4}$) | **45.6** | **71.2** | **52.7** | **42.2** | **64.8** |
| LSP (GCR, $d = 0$) | 43.8 | 69.8 | 50.2 | 40.0 | 64.8 |

Table 1: Performance of object detection task on the test set of our modified MNIST dataset.

matching cost that usually puts smaller importance on smaller objects (smaller bounding boxes). As a result, LSP improved the detection performance on small objects ($\text{AP}_{\text{S}}$) by +2.4 points over the baseline. A complete description of this task is provided in Appendix A.

---

[1]We observed that non-convergent trials demonstrated plateau loss curves within 300 iterations.

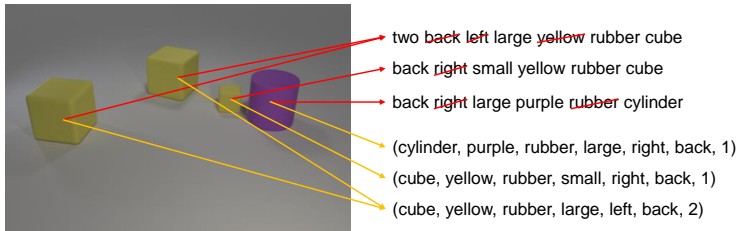

Figure 5: We repurposed CLEVR dataset [23] for object description task. The descriptions (red arrows) were from objects' attributes (yellow arrows) which came from the metadata. We randomly dropped attributes from descriptions to make the description generation task more challenging.

| Method | Precision | Recall | F1 |
|---|---|---|---|
| Concat | 0.931 | 0.910 | 0.920 |
| Ordered set | 0.957 | 0.526 | 0.679 |
| LSP (w/o GC) | 0.976 | 0.900 | 0.936 |
| LSP (GC) | **0.986** | 0.972 | **0.979** |
| LSP (GCR, $d = 10^{-3}$) | 0.983 | **0.975** | **0.979** |
| LSP (GCR, $d = 0$) | 0.983 | 0.972 | 0.978 |

Table 2: CLEVR object description generation task. Reported averages of three trials.

| Method | micro avg. BLEU | | |
| | $\hat{y} \to y$ | $y \to \hat{y}$ | hmean |
|---|---|---|---|
| RM+MCLN [26]* | 16.7 | 15.6 | 16.2 |
| Concat | 16.7 | 15.2 | 15.9 |
| Ordered set | **20.2** | 18.5 | 19.3 |
| LSP (GC) | 17.9 | 24.2 | 20.6 |
| LSP (GCR, $d = 10^{-3}$) | 19.1 | 24.6 | **21.5** |
| LSP (GCR, $d = 0$) | 18.9 | **24.7** | 21.4 |

Table 3: MIMIC-CXR report generation task. Reported averages of three trials except * which was run once.

## 5.3 CLEVR object description generation

Image captioning or paragraph captioning [24] can be considered a kind of set of texts prediction, yet are usually tackled as one long text. Set of texts is hard because teacher forcing does not work with Hungarian assignment. In this experiment and the next, we demonstrate that LSP enables set of texts prediction that leads to superior performances.

We re-purposed the CLEVR dataset [23], which was originally designed for visual reasoning, for image captioning. We selected CLEVR to represent a clean dataset which we know the ground truth exactly. Each image contains objects of different kinds (attributes) described by a text description derived from its attributes. Keywords were randomly dropped from the description to make the task more challenging. The final description was guaranteed to be unambiguous for each object to keep the task tractable. See Figure 5 for examples. We evaluated the models by the ratio of ground truth objects that were described by the model (recall) and the ratio of predicted descriptions that were supported by the ground truths (precision). We reported micro average precision, recall, and F1.

Image captioning is usually tackled by concatenating descriptions into a single caption. We imbued a deterministic alphabetical ordering[2] of descriptions to help the model learn. The model called **Concat** which resembles show-attend-tell [25] albeit with Transformer. A more reasonable approach is to describe each object description individually as a set. However, PIT does not facilitate teacher forcing during training. A practical approach is to turn a set into an **Ordered set** to circumvent the matching problem. Each prediction head in the model is responsible for generating the description of each object in an alphabetical order. The most appropriate approach should assume no fixed ordering as true set prediction which is possible with our **LSP** model. The results shown in Table 2 demonstrated strong performances from **LSP** while **Ordered set** lacked behind in recall. **Concat** was strong due to its fully autoregressive nature, yet not as strong as **LSP**. LSP results also improve with the addition of our proposed GC or GCR. Furthermore, modeling as sets allows the prediction heads to specialize. We observed this effect with the LSP model which was further described in Appendix Figure 10. A complete description of this task is provided in Appendix C.

## 5.4 Chest radiograph report generation

With the ability to do set of texts prediction, LSP has potentials for a challenging image captioning task such as chest radiograph report generation [26, 27] which has real-world applications [28]. We

---

[2]Alphabetical ordering was better than ordering by length in our preliminary experiments.

| X-ray | Ground truth | LSP | Ordered set | Concat |
|---|---|---|---|---|
|  | frontal and lateral views of the chest were obtained. there are streaky linear opacities at the lung bases which are likely due to atelectasis with chronic changes. no definite focal consolidation is seen. there is no pleural effusion or pneumothorax. no pneumothorax is seen. the aorta is calcified and tortuous. the cardiac silhouette is top normal to mildly enlarged. dual-lead left-sided pacemaker is seen with leads in the expected positions of the right atrium and right ventricle. chronic-appearing rib deformities on the right is again seen. | frontal and lateral views of the chest were obtained. there is a small left pleural effusion with overlying atelectasis. there is no focal consolidation, pleural effusion or pneumothorax. there is no pleural effusion or pneumothorax. the aorta is calcified and tortuous. the heart is mildly enlarged. a left-sided pacemaker is seen with leads in the expected position of the right atrium and right ventricle. the patient is status post median sternotomy and cabg. the lungs are otherwise clear. | pa and lateral views of the chest provided. there is no pneumothorax. no pneumothorax is seen. the heart is mildly enlarged. the patient is status post median sternotomy and cabg. the lungs are hyperinflated with flattening of the diaphragms, suggesting chronic obstructive pulmonary disease. | pa and lateral views of the chest provided. no large effusion or pneumothorax. the heart is mildly enlarged. midline sternotomy wires and mediastinal clips are noted. there is a left chest wall pacer device with lead tips extending to the region of the right atrium and right ventricle. there is mild pulmonary edema. no convincing signs of pneumonia. bony structures are intact. mediastinal contour is normal. no free air below the right hemidiaphragm. |

Table 4: A chest radiograph report generation example. Similar sentences were reordered and color coded for the ease of comparison. Hightlighted segments are major discrepancies from the ground truth pointed out by a radiologist.

used MIMIC-CXR dataset [16] which contains 377,110 chest x-rays with 227,835 reports from 65,379 patients with the average length of 50 words per report. It is important to note that set of texts prediction may not be much beneficial on shorter caption datasets such as MS-COCO [14]. MIMIC-CXR is considered noisy because multiple radiologists contributed to the dataset. The report may be incomplete and/or inconsistent depending on the writer. Also, the report is not entirely predictable because it usually refers to previous studies or to preconditions of the patient. We used a specific kind of BLEU score [29, 30] for evaluation which focuses on correctness and completeness. We calculated sentence-level BLEU scores from every source sentence to the highest BLEU target sentence. It was calculated both ways from predictions to ground truths $\hat{y} \to y$ and vice versa $y \to \hat{y}$. To get a single summary metric we compute the harmonic mean between $\hat{y} \to y$ and $y \to \hat{y}$. Note that we discarded all *blank* predictions before scoring. This score does not penalize duplicated predictions.

We included **Concat**, **Ordered set**, and Transformer with relational memory (**RM+MCLN**) [26], which is also a kind of **Concat**, as baselines against our **LSP** model. The three baselines followed the original ordering in the reports which was found to work better than alphabetical ordering. **Ordered set** and **LSP** modeled the task as a set of sentences. To better capture report diversity, we trained both models to always predict 10 sentences (from average 5.4 sentences per report). The same cannot be done with concatenation baselines. A representative example was shown in Table 4 (duplicate sentences were removed). Qualitatively, **Concat** generated a sound report. Like most radiologists, it mentioned just a few frequent negative findings. This shows that **Concat** was heavily biased by the imperfection of this dataset. **Ordered set** predicted the most duplicated sentences. Due to report diversity, the order of a particular finding sentence can differ between reports. This prevents **Ordered set** to specialize its prediction heads resulting in duplicated sentences being predicted. **LSP** predicted the most diverse sentences and was the best at capturing negative findings thanks to its head specialization. Quantitatively, Table 3 shows the superiority of both **Ordered set** and **LSP** models mainly due to their ability to over-predict. Although **Ordered set** has a slight edge on the $\hat{y} \to y$ metric, **LSP** has a substantial improvement in $y \to \hat{y}$ resulting in the best harmonic mean score. A complete description of this task and more prediction examples are provided in Appendix D.

# 6 Broader Impact and Limitation

LSP is applicable to all set prediction tasks as long as the set members are representable as latent vectors. Mature set prediction tasks like object detection are likely to receive only incremental improvements from LSP. However, LSP has larger implications on tasks that were previously hard to implement as set prediction including acoustic source separation. A questionable application like mass surveillance might be made possible by a practical acoustic source separation using LSP.

The proof of convergence (Section 4) relies on an assumption that both $s$'s and $g$'s respond to gradient updates exactly. This assumption is only satisfied without a neural network. Hence, we cannot mathematically guarantee the convergence in the general case.

Although LSP does away with the need for specifying distance metrics, it requires a *reasonable* encoder to be designed instead. One may argue that designing an encoder is not an easy task in some cases. One such example is the point cloud autoencoder task. Another example is the task of acoustic source separation where one wants to decompose a mixture of sounds. It might not be obvious what kind of encoder should be used in order to learn the latent information required in order to reconstruct each source.

In this paper, we investigated and designed encoders for a few tasks. One can use these as guidelines. However, the design of encoder in a completely different domain may require a non-trivial investment.

## 7 Conclusion

Set prediction requires a suitable distance metric that is also efficient to calculate. We proposed LSP as a potential answer to both criteria. We gave a theoretical model of LSP and showed its convergence properties under assumptions. This encourages usages in practical settings as we have shown with object detection and image captioning. LSP did away with the need for hand-crafted distance measures in object detection and made teacher forcing a viable option for set of text predictions. We envision that LSP will broaden the applicability of set prediction to other domains where a distance metric is hard to obtain or define.

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
