# Appendix

## Table of Contents

## A  Object detection

### A.1  Dataset

We used a modified MNIST dataset consisting of 5,000 training and 1,000 testing images. Each datapoint is a $160 \times 160$ image canvas containing multiple randomly placed digits from the MNIST dataset. To increase the task difficulty, each digit had to go through random image transformations

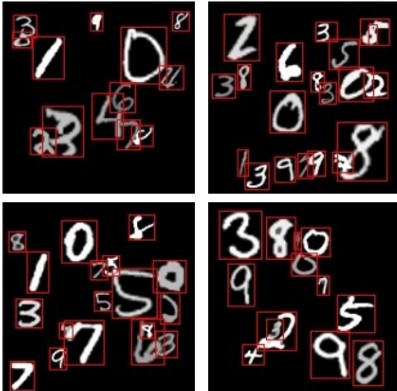

Figure 6: Example of datapoints in our object detection dataset and its ground truth (red boxes).

before being placed on the canvas. First, an image was randomly selected from the MNIST dataset and randomly resized to a square image with a size ranging from $16 \times 16$ to $64 \times 64$. After that, the resized image (digit) was further augmented by random brightness, contrast, Gaussian blur, opening, and closing morphological operations. Then, the augmented image (digit) was randomly placed on the canvas with a constraint preventing it from having an excessive amount of highly overlapped objects. We limited the maximum number of objects in a single image to 50. The digit placement was not allowed when the summation of gray-scale value (intensity) in the bounding box area is higher than 20,000. The bounding box of each digit in the MNIST dataset was obtained by finding the border of the largest connected component in the digit image. An example of the generated dataset is shown in Figure 6. The dataset will be released with the code base.

## A.2 Models

Every experiment was conducted using **DETR** with ResNet-50 [31] backbone, and 6 layers of both Transformer encoder and decoder with 256 hidden units and 8 attention heads. The backbone was ImageNet-pretrained. The size of class prediction heads was adjusted to match the number of classes in our dataset.

**LSP model** was based on **DETR** model with an extra encoder $\mathcal{E}$ component. We used an encoder $\mathcal{E}(\text{box}, \text{class}, \text{x})$ where $x$ is the average pooled features from the ResNet-50 backbone. The encoder had the following architecture:

$$a = \text{MLP}(\text{box})$$
$$b = \text{MLP}(\text{class})$$
$$c = \text{MLP}(\text{x})$$
$$g = \text{Linear}(\text{concat}(a, b, c))$$

Each $\text{MLP}$ is a three-layer MLP with 256 hidden units with layer norm and ReLU activation after each layer. $g$ has 256 units.

**Remarks on the LSP model.** The DETR model supervises on *all* layers of the Transformer decoder sharing the same prediction head. This aims to help training the deep architecture more effectively. Under the LSP terms, each intermediate state of the Transformer decoder layers is considered a set of $s$'s. There are *six* such layers hence six sets of $s$'s. We treated them individually in the experiments. That is we have six sets of $g$'s predicted by six different instances of encoders $\mathcal{E}$. This simplification disregards the fact that one set of $s$ may affect the other five, yet was found to work well in practice.

## A.3 Training details

We used the same set of training hyperparameters as DETR except for the number of training iterations, batch size, and augmentation strategy. Every model was trained for 100,000 iterations with the initial Transformer's learning rate of $10^{-4}$, and the backbone's learning rate of $10^{-5}$. The learning rate was divided by 10 after 75,000 iterations. Batch size of 32 and 8 were used for an

| Encoder $\mathcal{E}$ | AP | AP$_{50}$ | AP$_{75}$ | AP$_S$ | AP$_M$ |
|---|---|---|---|---|---|
| shared | 43.6 | 69.5 | 49.8 | 40.2 | 63.8 |
| separated | **45.6** | **71.2** | **52.7** | **42.2** | **64.8** |

Table 5: Comparing shared vs. separated encoders $\mathcal{E}$ in the object detection task.

| Method | Batch size | AP | AP$_{50}$ | AP$_{75}$ | AP$_S$ | AP$_M$ |
|---|---|---|---|---|---|---|
| DETR [2] | 8 | 33.5 | 66.3 | 30.0 | 30.2 | 51.1 |
| | 16 | 41.7 | 70.7 | 45.4 | 38.4 | 61.6 |
| | 32 | **43.5** | **71.0** | **49.1** | **39.8** | **64.5** |
| | 64 | 41.4 | 68.5 | 46.2 | 37.5 | 63.0 |
| Ours | 8 | **45.6** | **71.2** | **52.7** | **42.2** | **64.8** |
| | 16 | 40.1 | 66.0 | 44.8 | 36.2 | 62.7 |
| | 32 | 38.1 | 63.7 | 42.0 | 34.5 | 56.9 |

Table 6: The effect of different batch size on model performance.

original DETR and the DETR with LSP respectively. Training images were augmented using random horizontal flip, random brightness, random contrast, and random Gaussian blur. The training image resolution was set to $160 \times 160$ pixels.

### A.4   Ablation studies

All proposed model used in this section is DETR with LSP (GCR), with $d = 10^{-4}$. We used the same training schedule as the main experiments.

#### A.4.1   Choice of encoder $\mathcal{E}$

We evaluate the necessity of having a different encoder $\mathcal{E}$ for each Transformer decoder by comparing it with the one which has a single shared encoder $\mathcal{E}$ for every Transformer decoder. Table 5 shows that although having multiple sets of $s$ may affect the others, having different $\mathcal{E}$ outperformed a single shared one. Nevertheless, having shared $\mathcal{E}$ still achieved a competitive performance to the DETR baseline.

#### A.4.2   Batch size

We evaluate the effect of different training batch sizes of the proposed method. Table 6 shows that our method achieved the best performance at the batch size of 8. Surprisingly, the performance dropped drastically as the batch size increases which is opposite from the DETR baseline. The cause for this problem is unclear and is open for future research.

### A.5   Convergence speed

Figure 7 shows a training progression plot of LSP (GCR) with different $d$'s against the DETR baseline. Both methods achieved convergence without much instability. However, we observed slightly worse small object localization performance from the DETR baseline compared to that of learned distance function from LSP.

### A.6   Qualitative results

Figure 8 shows a qualitative result of DETR and our method on our generated object detection dataset. The images in the figure are randomly selected from the test set.

## B   MNIST Point Cloud Autoencoding

In this section, we aim to show that LSP is competitive on the often used set prediction task, namely point cloud autoencoding on the MNIST dataset [12]. We provided a comparison of our method against DSPN [12], and TSPN [1]. In contrast to prior works which use the chamfer loss, we performed comparison based on the Hungarian assignment.

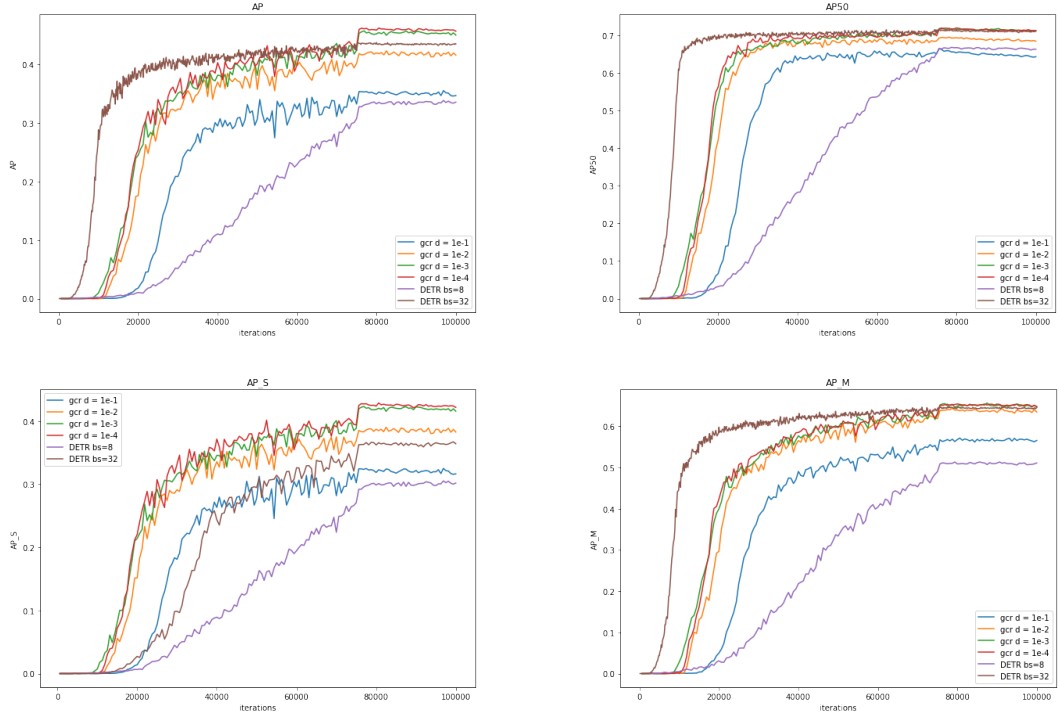

Figure 7: Convergence plot of GCR against the DETR baseline on the object detection task.

## B.1 Model

We followed DSPN and TSPN model architectures by using a 3-layer MLP with FSPool [32] for the set encoders. For LSP, we augmented the TSPN architecture with an encoder, which is simply a linear layer from a tuple of (x, y, presence) to a 256-sized vector. For both the TSPN and LSP variants, we used the learned embeddings instead of Gaussian random vectors which were found to work better for Hungarian matching. In addition, both predicted each element's presence to derive the set cardinality instead of using an MLP to predict the number of set cardinality explicitly.

## B.2 Experimental setup

We followed DSPN and TSPN point cloud MNIST dataset, but limited the maximum number of points to 150. All experiments were run for 50 epochs. Each performance number was the minimum of the run. We varied the "hidden dimension" of the encoder for DSPN to scale it up for a comparable parameter count. DSPN used a learning rate of 0.01 (following DSPN, grid searched from [0.01, 0.001, 0.0001]) while the others used a learning rate of 0.0001. We used batch normalization in the set encoder for faster convergence.

## B.3 Results

Table 7 shows a result of LSP aganist the DSPN and TSPN baseline. DSPN lags behind both TSPN and LSP by a large margin both quantitatively and qualitatively (Figure 9). DSPN also did not scale up well with wider models. TSPN performed better than LSP in this experiment, yet qualitatively hard to perceive the differences. We want to point out that this is possibly a task where designing a good encoder is harder than designing a good distance metric since each set element is simply (x, y, presence).

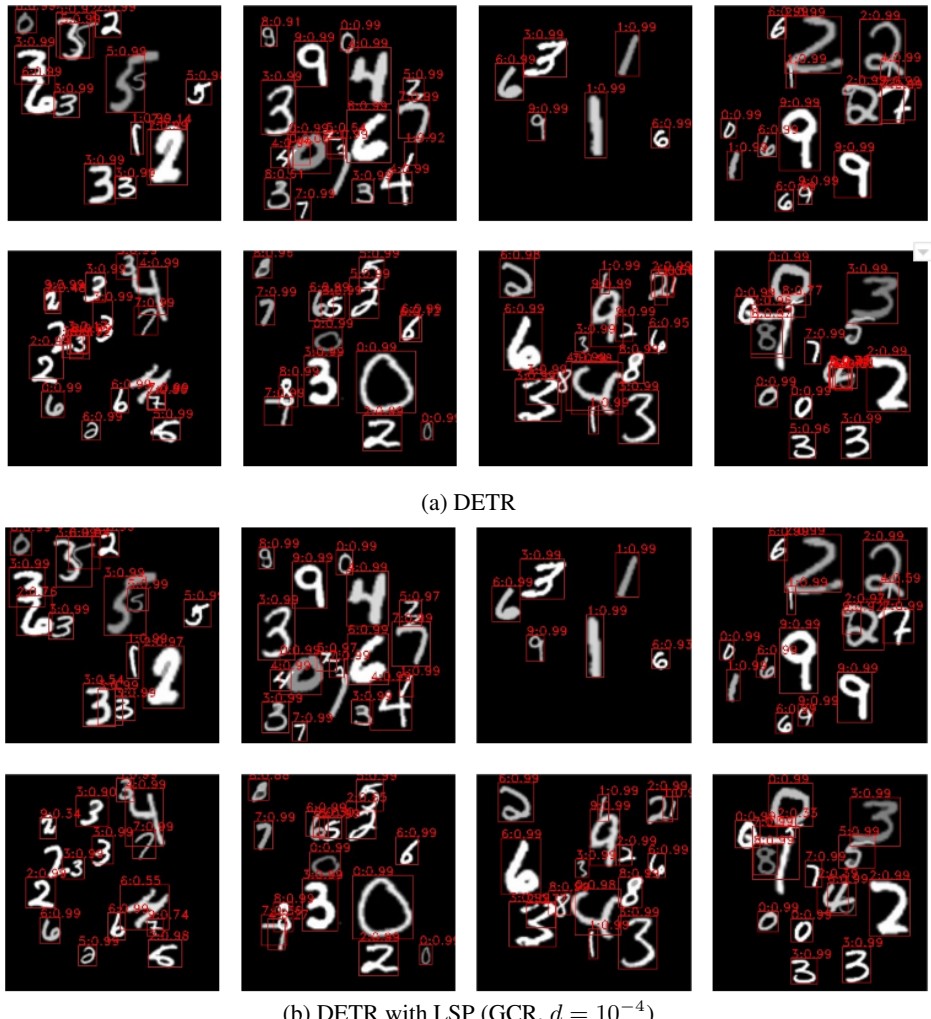

(a) DETR

(b) DETR with LSP (GCR, $d = 10^{-4}$)

Figure 8: Qualitative result of DETR and DETR with LSP on our object detection dataset.



(a) DSPN 256     (b) DSPN 512     (c) DSPN 1024     (d) TSPN     (e) LSP

Figure 9: The qualitative comparison between reconstructed images from different models for the point cloud autoencoding task. DSPN's performed poorly qualitatively compared to TSPN and LSP.

# C   CLEVR object description generation

## C.1   Dataset

We constructed the attribute prediction dataset from images and metadata of CLEVR dataset [23]. The dataset released with 70,000/10,000 train/val scenes. Note that we used the *val* scenes as our test dataset and split the *train* into 60,000/10,000 for training and development purposes. There are seven attributes: shape, color, material, size, left/right, front/back, object count. Four of them

| Model | Hidden dim. | #Params (M) | Chamfer L2 distance |
|---|---|---|---|
| DSPN | 256 | 0.14 | 4.99 ± 0.37 |
| DSPN | 512 | 0.40 | 4.52 ± 0.23 |
| DSPN* | 1024 | 1.33 | 4.42 ± 0.01 |
| TSPN | 256 | 1.50 | **3.78 ± 0.01** |
| LSP (GCR, $d = 0.001$) | 256 | 1.50 | 3.90 ± 0.02 |

Table 7: The performance comparison for test set of MNIST point cloud autoencoder task. The symbol ± represents one standard deviation of 3 different random seeds, except DSPN* that runs with 2 seeds.

were obtained from the dataset's metadata. The additional left/right and front/back attributes were calculated from the object's pixel-wise quadrant. The object count attribute ($\leq 3$) is the number of objects that share the same attributes. The object description was generated from these attributes with random corruption. All attributes except *shape* were dropped with 50% chance. Note that the corruption always retained the unambiguity of the description.

## C.2 Models

All models used byte-level byte pair encoding same as RoBERTa [33] via Huggingface [34]. We used ResNet-34 [31] pretrained on Imagenet as the backbones for all models.

**Concat model** is a model where the input was the concatenation of all object descriptions separated by "<sep>" tokens. It has the following architecture.

$$\textbf{feat} = \text{Conv1}(\text{ResNet34}(x))$$

$$\hat{\textbf{Y}} = \text{TransformerDecoder}(\textbf{Y}, \textbf{feat})$$

where TransformerDecoder is a 3-layer Transformer decoder with 256 hidden units and 4 attention head connected to a linear layer for predicting the softmax distribution of the vocabularies, and $\textbf{Y}$ is supplied as a teacher forcing signal. At the evaluation time, the network was greedily decoded.

**Ordered set model** predicts an alphabetically sorted list of object descriptions. It has the following architecture.

$$\textbf{feat} = \text{Conv1}(\text{ResNet34}(x))$$

$$\textbf{R} = \text{TransformerDecoder}(\textbf{seed}, \textbf{feat})$$

$$\hat{\textbf{Y}} = \mathcal{D}(\textbf{R} + \textbf{Y}, \text{repeat}(\textbf{feat}))$$

where TransformerDecoder is a 3-layer Transformer decoder with 256 hidden units and 4 attention heads, **seed** is fixed sinusoidal vectors as queries for set elements, $\mathcal{D}$ is 3-layer Transformer decoder for generating object descriptions with 256 hidden units and 4 heads connected to a linear layer for predicting the softmax distribution of the vocabularies, and $\textbf{Y}$ is supplied as a teacher forcing signal. $\textbf{R}$ is *added* to the input of $\mathcal{D}$ to dictate the topic about which it generates. The network always predicted $K$ sentences ($K = 10$). We padded the ground truths ($N$ sentences) with blank lines until they have $K$ sentences. At the evaluation time, the network was greedily decoded, and all the blank lines were removed.

**LSP model** predicts a set of object descriptions. It shared the above architecture with an addition of an encoder $\mathcal{E}$ as follows

$$\textbf{B} = \text{TransformerDecoder}(\textbf{Y}, \text{repeat}(\textbf{feat}))$$

where TransformerDecoder is a 3-layer Transformer decoder with 256 hidden units and 4 attention heads. It performs cross-attention between the ground truth $\textbf{Y}$ and the image feature $\textbf{feat}$. In summary, object descriptions $\textbf{Y}$ were encoded as $\textbf{B}$. Then, the Hungarian algorithm was performed to find the minimum assignment $\pi$ between $\textbf{R}$ and $\textbf{B}$ under Euclidean distance. The decoding step of this network became $\hat{\textbf{Y}}_\pi = \mathcal{D}(\textbf{R}_\pi + \textbf{Y}, \text{repeat}(\textbf{feat}))$ following the seq of text model's notation. Finally, the loss function was calculated directly as $\mathcal{L}_{\text{task}}(\hat{\textbf{Y}}_\pi, \textbf{Y})$. Like the seq of text model, the ground truths $\textbf{Y}$ were padded by blank lines to have the total of $K$ sentences. All the blank lines were removed at inference time.

## C.3 Training details

The model sizes and other hyperparameters were not heavily tuned. Our goal is to show the improvement from modeling the task as set prediction. **Optimization.** Adam with learning rate $10^{-4}$. Batch size 64. The learning rate was reduced by 5 after the validation loss of $\mathcal{L}_{\text{task}}$ is not reducing for 2 epochs. When the learning rate was below $10^{-6}$, the training stopped. **Augmentation.** Besides resizing the image to $256 \times 256$, there is no other augmentation.

## C.4 Results with standard deviations

We reported averages of three trials. $\pm$ denotes single standard deviation.

| Method | Precision | Recall | F1 |
|---|---|---|---|
| Concat | $0.931 \pm 0.02$ | $0.910 \pm 0.02$ | $0.920 \pm 0.01$ |
| Ordered set | $0.957 \pm 0.01$ | $0.526 \pm 0.02$ | $0.679 \pm 0.02$ |
| LSP (GC) | $\mathbf{0.986} \pm 0.01$ | $0.972 \pm 0.01$ | $\mathbf{0.979} \pm 0.01$ |
| LSP (GCR, $d = 10^{-1}$) | $0.983 \pm 0.01$ | $0.970 \pm 0.01$ | $0.977 \pm 0.01$ |
| LSP (GCR, $d = 10^{-2}$) | $0.979 \pm 0.01$ | $0.957 \pm 0.02$ | $0.968 \pm 0.01$ |
| LSP (GCR, $d = 10^{-3}$) | $0.983 \pm 0.01$ | $\mathbf{0.975} \pm 0.01$ | $\mathbf{0.979} \pm 0.01$ |
| LSP (GCR, $d = 10^{-4}$) | $0.984 \pm 0.01$ | $0.973 \pm 0.02$ | $0.978 \pm 0.02$ |
| LSP (GCR, $d = 0$) | $0.983 \pm 0.01$ | $0.972 \pm 0.02$ | $0.978 \pm 0.01$ |

## C.5 Effect of hyperparameter in asymmetric latent loss

In this section, we studied the effect of hyperparameter $\beta$ and $\gamma$ of the asymmetric latent loss ($\mathcal{L}_{\text{latent}}$) by fixing the value of $\gamma$ to 1 and adjusted the $\beta$. The table below shows us that $\beta = 0.1$ yielded the best performance. Nevertheless, given the variances, there was no significant performance change over different $\beta$.

| Model | Precision | Recall | F1 |
|---|---|---|---|
| GC ($\beta = 0$) | $0.979 \pm 0.01$ | $0.970 \pm 0.02$ | $0.975 \pm 0.01$ |
| GC ($\beta = 0.1$) | $\mathbf{0.989} \pm 0.01$ | $\mathbf{0.979} \pm 0.01$ | $\mathbf{0.984} \pm 0.01$ |
| GC ($\beta = 0.2$) | $0.987 \pm 0.01$ | $0.976 \pm 0.02$ | $0.982 \pm 0.01$ |
| GC ($\beta = 0.5$) | $0.980 \pm 0.01$ | $0.972 \pm 0.01$ | $0.976 \pm 0.01$ |
| GC ($\beta = 1$) | $0.983 \pm 0.01$ | $0.966 \pm 0.02$ | $0.974 \pm 0.02$ |

## C.6 Effect of GCR's $d$

In Section C.4. We saw no real performance differences between GC and GCR and GCR with different $d$'s. This task represents a non-trivial set prediction yet has clean labels and few set members. Under this scenario, the choice of GC or GCR did not really matter.

## C.7 Head specialization

We observed prediction head specialization from the **LSP** model both in location specialization (Figure 10a) and shape specialization (Figure 10b).

## C.8 Convergence speed comparison

We depicted the speed of validation metrics over training epochs between Ordered Set, Concat, and LSP (GC) in Figure 11. Two baselines converged faster than LSP but to worse solutions. Since these methods did not converge to solutions of the same quality, it was unfair to compare the convergence time directly. However, at any point in time, LSP was either on par or better with the other methods, showing training stability. Noted that it is not a perfect comparison because the baselines are not set prediction methods.

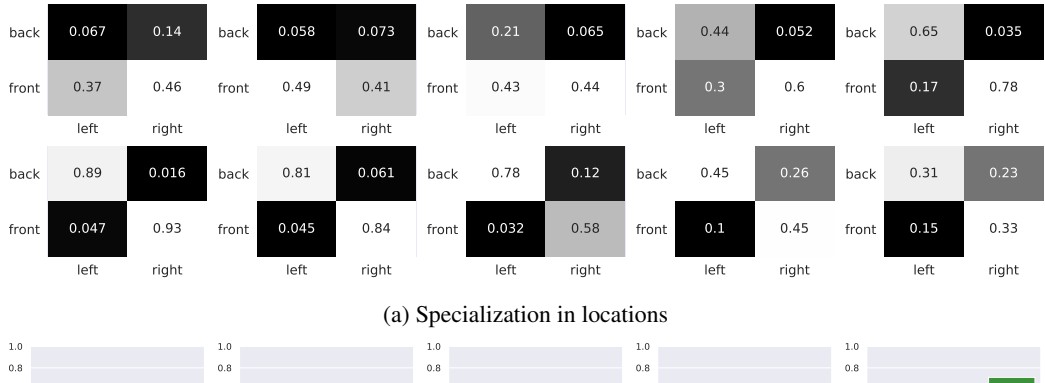

(a) Specialization in locations

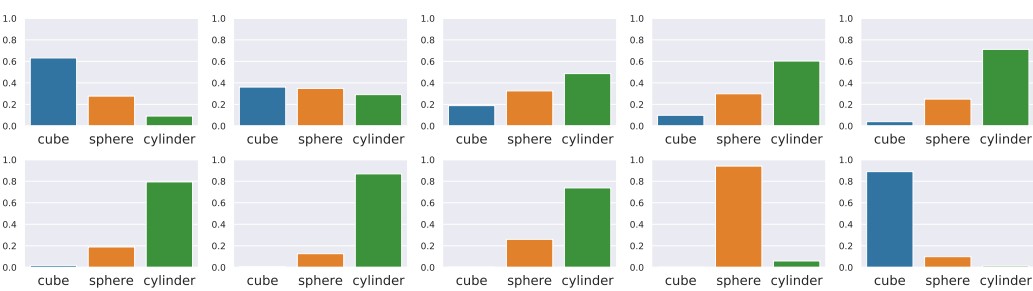

(b) Specialization in shapes

Figure 10: All 10-head specialization of the LSP model on CLEVR object description task.

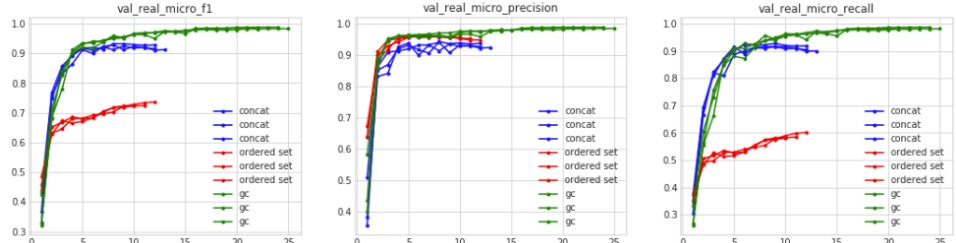

Figure 11: The comparison between validation performance progresses of different methods for CLEVR dataset. Different lines are different random seeds.

# D    Chest radiograph report generation

## D.1    Dataset

We used MIMIC-CXR dataset [16] containing 377,110 chest x-rays with 227,835 reports from 65,379 patients. We selected only reports with apparent "FINDING" keyword and extracted only the finding section with regular expression to focus on a specific part on the report that is most predictable from a chest x-ray. This resulted in 149,766 reports. From these, we selected only "frontal" images (PA or AP). Finally, we have 160,291 images and 143,778 reports which were split into train/val/test as 112,025/15,994/32,272 images and 100,531/14,391/28,856 reports respectively. We broke a document into sentences with Spacy [35].

## D.2    Models

All models used byte-level byte pair encoding same as RoBERTa [33] via Huggingface [34]. We used ResNet-34 [31] pretrained on Imagenet as the backbones for all models.

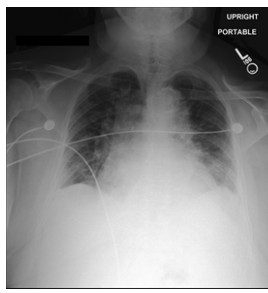

FINDINGS: In comparison to study performed on of ___ there is new mild pulmonary edema with small bilateral pleural effusions. Lung volumes have decreased with crowding of vasculature. No pneumothorax. Severe cardiomegaly is likely accentuated due to low lung volumes and patient positioning.
IMPRESSION: 1. New mild pulmonary edema with persistent small bilateral pleural effusions. 2. Severe cardiomegaly is likely accentuated due to low lung volumes and patient positioning.

Figure 12: A sample report from MIMIC-CXR dataset [16]. A report usually contains header section, finding section, and impression section. We extracted only the finding section with regular expression (highlighted in yellow).

**Concat model** is a model where the input was the concatenation of finding sentences in the report separated by "<sep>" tokens. It has the following architecture.

$$\mathbf{feat} = \mathrm{Conv1}(\mathrm{ResNet34}(x))$$
$$\mathbf{feat} = \mathrm{TransformerEncoder}(\mathbf{feat} + \text{2D position encoding})$$
$$\hat{\mathbf{Y}} = \mathrm{TransformerDecoder}(\mathbf{Y}, \mathbf{feat})$$

where TransformerEncoder is a 3-layer Transformer encoder with 256 hidden units and 4 attention heads, the position encoding is 2D sinusoidal features following [2], TransformerDecoder is a 3-layer Transformer decoder with 256 hidden units and 4 attention heads connected to a linear layer for predicting the softmax distribution of the vocabularies, and $\mathbf{Y}$ is supplied as a teacher forcing signal. At the evaluation time, the network was greedily decoded.

**Ordered set model** predicts an alphabetically sorted list of finding sentences. It has the following architecture.

$$\mathbf{feat} = \mathrm{Conv1}(\mathrm{ResNet34}(x))$$
$$\mathbf{feat} = \mathrm{TransformerEncoder}(\mathbf{feat} + \text{2D position encoding})$$
$$\mathbf{R} = \mathrm{TransformerDecoder}(\mathbf{seed}, \mathbf{feat})$$
$$\hat{\mathbf{Y}} = \mathcal{D}(\mathbf{R} + \mathbf{Y}, \mathrm{repeat}(\mathbf{feat}))$$

where TransformerEncoder is a 3-layer Transformer encoder with 256 hidden units and 4 attention heads, the position encoding is 2D sinusoidal features following [2], TransformerDecoder is a 3-layer Transformer decoder with 256 hidden units and 4 attention heads, **seed** is fixed sinusoidal vectors as queries for set elements, $\mathcal{D}$ is 3-layer Transformer decoder for generating object descriptions with 256 hidden units and 4 heads connected to a linear layer for predicting the softmax distribution of the vocabularies, and $\mathbf{Y}$ is supplied as a teacher forcing signal. $\mathbf{R}$ is *added* to the input of $\mathcal{D}$ to dictate the topic about which it generates. The network always predicted $K$ sentences ($K = 10$). We padded the ground truths ($N$ sentences) with blank lines until they have $K$ sentences. The losses on these padded blank lines were *zero* out to allow the model to always predict $K$ non-blank sentences. At the evaluation time, the network was greedily decoded.

**LSP model** predicts a set of finding sentences. It shared the above architecture with an addition of an encoder $\mathcal{E}$ as follows

$$\mathbf{B} = \mathrm{TransformerDecoder}(\mathbf{Y}, \mathrm{repeat}(\mathbf{feat}))$$

where TransformerDecoder is a 3-layer Transformer decoder with 256 hidden units and 4 attention heads. It performs cross-attention between the ground truth $\mathbf{Y}$ and the image feature $\mathbf{feat}$. In summary, object descriptions $\mathbf{Y}$ were encoded as $\mathbf{B}$. Then, the Hungarian algorithm was performed to find the minimum assignment $\pi$ between $\mathbf{R}$ and $\mathbf{B}$ under Euclidean distance. The decoding step of this network became $\hat{\mathbf{Y}}_\pi = \mathcal{D}(\mathbf{R}_\pi + \mathbf{Y}, \mathrm{repeat}(\mathbf{feat}))$ following the sequence of text model's notation. Finally, the loss function was calculated directly as $\mathcal{L}_{\mathrm{task}}(\hat{\mathbf{Y}}_\pi, \mathbf{Y})$. This model always predicted $K = 10$ sentences. However, we *did not* pad $\mathbf{Y}$ with blank sentences. That is the Hungarian match of $N$ out of $K$ sentences. We found this to work better than that with blank sentence padding.

**RM+MCLM [26]** also takes the concatenation of finding sentences in the report. We used the official implementation from https://github.com/cuhksz-nlp/R2Gen. We ran it with our dataset using batch size 64 and our augmentation scheme. We kept the rest of the settings original apart from that of the other models.

## D.3 Training details

The model sizes and other hyperparameters were not heavily tuned. Our goal is to show the improvement from modeling the task as set prediction. **Optimization.** Adam with learning rate $10^{-4}$. Batch size 64. The learning rate was reduced by 5 after the validation loss of $\mathcal{L}_{\text{task}}$ is not reducing for 2 epochs. When the learning rate was below $10^{-6}$, the training stopped. **Augmentation.** Random rotation up to 90 degrees, random horizontal flip, random contrast and brightness in range (0.5, 1.5), random crop with random size in range (0.7, 1.0), and random aspect ratio from 4:3 to 3:4. Images were resized to $256 \times 256$ pixels.

## D.4 Results with standard deviations

We reported averages of three trials except $^*$ which was run once. $\pm$ denotes single standard deviation.

| Method | micro avg. BLEU | | |
| --- | --- | --- | --- |
| | $\hat{y} \to y$ | $y \to \hat{y}$ | hmean |
| RM+MCLN [26]$^*$ | 16.7 | 15.6 | 16.2 |
| Concat | $16.7 \pm 0.1$ | $15.2 \pm 0.1$ | $15.9 \pm 0.1$ |
| Ordered set | $\mathbf{20.2} \pm 0.2$ | $18.5 \pm 0.2$ | $19.3 \pm 0.2$ |
| LSP (GC) | $17.9 \pm 0.5$ | $24.2 \pm 0.5$ | $20.6 \pm 0.3$ |
| LSP (GCR, $d = 10^{-1}$) | $19.1 \pm 0.5$ | $24.5 \pm 0.5$ | $\mathbf{21.5} \pm 0.5$ |
| LSP (GCR, $d = 10^{-2}$) | $18.7 \pm 0.8$ | $24.6 \pm 0.5$ | $21.3 \pm 0.7$ |
| LSP (GCR, $d = 10^{-3}$) | $19.1 \pm 0.2$ | $24.6 \pm 0.1$ | $\mathbf{21.5} \pm 0.1$ |
| LSP (GCR, $d = 10^{-4}$) | $18.6 \pm 0.7$ | $24.3 \pm 0.2$ | $21.1 \pm 0.5$ |
| LSP (GCR, $d = 0$) | $18.9 \pm 0.8$ | $\mathbf{24.7} \pm 0.4$ | $21.4 \pm 0.7$ |

## D.5 Effect of GCR's $d$

In Section D.4, we observed high $y \to \hat{y}$ across GC and GCR with all $d$'s. The difference was in $\hat{y} \to y$ where GCR excelled. We observed no real differences between different $d$'s of GCR. We began to saw superior performances from GCR over GC in this difficult image captioning task with noisy labels.

## D.6 Prediction samples

We selected a few reports to qualitatively evaluate the models. Duplicate sentences were dropped in the post process. We reordered and color-coded sentences related to abnormalities by hand to the best of our non-expert ability. Finally, we asked a radiologist for opinions on these reports and highlighted areas that significantly deviated from the ground truth. The radiologist was also asked to select the best model from each example. For example number 1-3 the LSP is preferred, while the best choice for number 4 is inconclusive.

In general, there are abnormalities such as opacity, consolidation, and nodule that were not frequently recognized by the models. The models were quite able to recognize abnormalities like pleural effusions, cardiomegaly, and devices including lines and pacemaker. However, we observed better prediction coverage from the LSP model than its peers especially on negative abnormality predictions. In our opinion, this task is not yet solved. It is questionable how much more we can expect models to learn from this noisy and partially predictable dataset.

## Report generation example 1 (curated)

| X-ray | Ground truth | LSP | Ordered set | Concat |
|---|---|---|---|---|
|  | frontal and lateral views of the chest were obtained. there are streaky linear opacities at the lung bases which are likely due to atelectasis with chronic changes. no definite focal consolidation is seen. there is no pleural effusion or pneumothorax. no pneumothorax is seen. the aorta is calcified and tortuous. the cardiac silhouette is top normal to mildly enlarged. dual-lead left-sided pacemaker is seen with leads in the expected positions of the right atrium and right ventricle. chronic-appearing rib deformities on the right is again seen. | frontal and lateral views of the chest were obtained. there is a small left pleural effusion with overlying atelectasis. there is no focal consolidation, pleural effusion or pneumothorax. there is no pleural effusion or pneumothorax. the aorta is calcified and tortuous. the heart is mildly enlarged. a left-sided pacemaker is seen with leads in the expected position of the right atrium and right ventricle. the patient is status post median sternotomy and cabg. the lungs are otherwise clear. | pa and lateral views of the chest provided. there is no pneumothorax. no pneumothorax is seen. the heart is mildly enlarged. the patient is status post median sternotomy and cabg. the lungs are hyperinflated with flattening of the diaphragms, suggesting chronic obstructive pulmonary disease. | pa and lateral views of the chest provided. no large effusion or pneumothorax. the heart is mildly enlarged. midline sternotomy wires and mediastinal clips are noted. there is a left chest wall pacer device with lead tips extending to the region of the right atrium and right ventricle. there is mild pulmonary edema. no convincing signs of pneumonia. bony structures are intact. mediastinal contour is normal. no free air below the right hemidiaphragm. |

## Report generation example 2 (curated)

| X-ray | Ground truth | LSP | Ordered set | Concat |
|---|---|---|---|---|
|  | the atient is status post coronary artery bypass graft surgery. the heart is mildly enlarged. there is a large hiatal hernia with an air-fluid level. otherwise, the mediastinal and hilar contours are unremarkable. the lungs appear clear. the chest is hyperinflated. there is no pleural effusion or pneumothorax. bony structures are unremarkable. | the patient is status post median sternotomy and cabg. the heart is mildly enlarged. there is a large hiatal hernia with an air-fluid level. the mediastinal and hilar contours are unremarkable. the lungs are clear. there is no pleural effusion or pneumothorax. there is no focal consolidation, pleural effusion or pneumothorax. no acute osseous abnormalities. frontal and lateral views of the chest were obtained. | the patient is status post median sternotomy and cabg. the cardiac silhouette is top normal to mildly enlarged. the lungs are clear. no acute osseous abnormalities. no acute osseous abnormalities are detected. pa and lateral views of the chest were provided. no free air below the right hemidiaphragm. | midline sternotomy wires and mediastinal clips are again noted. the cardiomediastinal silhouette is normal. there is a large retrocardiac opacity containing an air-fluid level concerning for a large hiatal hernia. there is no focal consolidation, effusion, or pneumothorax. imaged osseous structures are intact. pa and lateral views of the chest provided. no free air below the right hemidiaphragm is seen. |

## Report generation example 3 (random)

| X-ray | Ground truth | LSP | Ordered set | Concat |
|---|---|---|---|---|
|  | pa and lateral views of the chest provided. left chest wall dual lead pacer is again noted with leads extending the region the right atrium and right ventricle. the heart is mildly enlarged. no convincing signs of edema or pneumonia. mediastinal contour is normal. bony structures are intact. mild hilar congestion is suspected. | ap and lateral views of the chest. left-sided dual-chamber pacemaker device is noted with leads terminating in the right atrium and right ventricle. the heart is mildly enlarged. the mediastinal and hilar contours are unchanged. no acute osseous abnormalities. there is mild pulmonary vascular congestion. no pleural effusion or pneumothorax is seen. there is no pleural effusion or pneumothorax. there is no focal consolidation, pleural effusion or pneumothorax. the lungs are clear. | left-sided dual-chamber pacemaker device is noted with leads terminating in the right atrium and right ventricle. the heart is mildly enlarged. the cardiac silhouette is mildly enlarged. no acute osseous abnormality is detected. the lungs are clear of focal consolidation or effusion. the lungs are clear without focal consolidation, effusion, or pneumothorax. no focal consolidation, pleural effusion or pneumothorax is present. no free air below the right hemidiaphragm. | left-sided dual-chamber pacemaker device is noted with leads terminating in the right atrium and right ventricle. mild cardiomegaly is unchanged. there is no pulmonary edema. the mediastinal and hilar contours are similar. there are no acute osseous abnormalities. no focal consolidation, pleural effusion or pneumothorax is present. |

**Report generation example 4 (random)**

| X-ray | Ground truth | LSP | Ordered set | Concat |
|---|---|---|---|---|
|  | a portable view of the chest demonstrates continous improved aeration of the left mid and upper lung. a moderate left pleural effusion remains. small right pleural effusion is stable. the right lung is grossly clear. a left pigtail catheter and right picc are unchanged in position. there is no pneumothorax. | as compared to the previous radiograph, the patient has been extubated. there is a large left pleural effusion with associated atelectasis. the right lung is essentially clear. the tip of the right picc line projects over the mid svc. the monitoring and support devices are constant. there is no pneumothorax. no pneumothorax. the size of the cardiac silhouette is unchanged. the heart is enlarged. there is mild pulmonary edema. | as compared to the previous radiograph, the patient has received a new right-sided picc line. there is a right-sided picc line with distal lead tip in the distal svc. there is no pneumothorax. | the pre-existing left pleural effusion has slightly increased in extent. as compared to the previous radiograph, the patient has received a right-sided picc line. the course of the line is unremarkable, the tip of the line projects over the mid svc. the tip of the line projects over the mid svc. there is no evidence of complications, notably no pneumothorax. unchanged appearance of the right lung. unchanged appearance of the cardiac silhouette. |

# E    Proof of convergence

We begin by showing the proof for the simplest situation where **gradient cloning** is used without the **rejection** mechanism and $\beta$ is set to 0. This will illustrate the intuitions behind the convergence of latent set prediction, which can be modified to accommodate the **rejection** mechanism and other values of $\beta$ later on.

## E.1    Recall of the setup

In the deep set prediction setting, we wish to train a set predictor which generates latent vectors $s_i \in \mathbb{R}^C$ to match a set of targets $y_i$ in the output space, which are designated by the generated guiding vectors $g_i \in \mathbb{R}^C$. With a permutation $\pi$, we introduce the main task's loss

$$\mathcal{L}_{\text{task}} = \sum_i l_{\text{task}}(s_{\pi(i)}, y_i).$$

Note that the permutation $\pi$ can change over the course of model training and a **switch** (Figure 2) is said to occur when $\pi^{(t+1)} \neq \pi^{(t)}$. To discourage **switch**, we define the following latent loss with squared Euclidean distance to drive $g_i$ toward $s_{\pi(i)}$

$$\mathcal{L}_{\text{latent}} = \sum_i l_{\text{latent}}(s_{\pi(i)}, g_i) = \sum_i \frac{1}{2}\|s_{\pi(i)} - g_i\|_2^2.$$

The training at each time point $t + 1$ consists of two steps. First, the permutation $\pi$ is updated according to the values of $s_{\pi(i)}^{(t)}$'s and $g_i^{(t)}$'s from the previous time point

$$\pi^{(t+1)} = \underset{\pi' \in \mathcal{P}}{\operatorname{argmin}} \sum_i \|s_{\pi'(i)}^{(t)} - g_i^{(t)}\|_2, \text{ where } \mathcal{P} \text{ is the set of all permutations.} \tag{4}$$

Then, the values of $s_{\pi(i)}$'s and $g_i$'s are updated based on the gradients from $\mathcal{L}_{\text{task}}$ and $\mathcal{L}_{\text{latent}}$ with step size $\eta$ and latent loss strength $\gamma$

$$\begin{aligned}
s_{\pi^{(t+1)}(i)}^{(t+1)} &= s_{\pi^{(t+1)}(i)}^{(t)} - \eta \nabla_s l_{\text{task}}(s_{\pi^{(t+1)}(i)}^{(t)}, y_i) \\
g_i^{(t+1)} &= g_i^{(t)} - \eta \left( \nabla_s l_{\text{task}}(s_{\pi^{(t+1)}(i)}^{(t)}, y_i) + \gamma \nabla_g l_{\text{latent}}(s_{\pi^{(t+1)}(i)}^{(t)}, g_i^{(t)}) \right).
\end{aligned} \tag{5}$$

This is the **gradient cloning** technique with a special case of **asymmetric latent loss** (as in (2) where $\beta = 0$).

Even though the set matching step may increase task loss through the switch from $s_{\pi(i)}^{(t)}$ to $s_{\pi(i)}^{(t+1)}$, it turns out that sufficient conditions for the convergence of LSP are the same as those needed in a typical proof of convergence of gradient descent [21]. Namely, we assume that $l_{\text{task}}(\cdot, y)$ is **L-smooth** and satisfies the **Polyak-Lojasiewicz** condition.

The **L-smooth** assumption states that there is a constant $L \in \mathbb{R}^+$ such that for any $x$ and $y$

$$L \cdot \|x - y\|_2 \geq \|\nabla_s l_{\text{task}}(x, y) - \nabla_s l_{\text{task}}(y, y)\|_2, \tag{6}$$

which also implies (Lemma 1.2 in [36])

$$\langle \nabla_s l_{\text{task}}(x, y), y - x \rangle + \frac{L}{2}\|y - x\|_2^2 \geq l_{\text{task}}(y, y) - l_{\text{task}}(x, y). \tag{7}$$

The **Polyak-Lojasiewicz** condition guarantees that there is a constant $\mu \in \mathbb{R}^+$ such that for any $x$

$$\frac{1}{2}\|\nabla_s l_{\text{task}}(x, y)\|^2 \geq \mu \cdot (l_{\text{task}}(x, y) - l_{\text{task}}(x^*, y)), \text{ where } x^* \text{ is where the function reaches its minimum} \tag{8}$$

## E.2 Exponential decay of the switch distance

In this section, we show that our training dynamics effectively drive $g_i$'s toward $s_{\pi(i)}$'s so strongly that the distance between a **switch** at time $t$ decays exponentially.

First, we show that the total distance between respective $g$'s and $s$'s decay exponentially.

**Lemma E.1.** *For* $0 < \eta\gamma \leq 1$,

$$\sum_i \|s_{\pi^{(t)}(i)}^{(t)} - g_i^{(t)}\|_2 \leq (1 - \eta\gamma)^t \sum_i \|s_{\pi^{(0)}(i)}^{(0)} - g^{(0)}\|_2.$$

*Proof.* From the gradient descent update formula for $s_{\pi^{(t+1)}(i)}^{(t+1)}$ and $g_i^{(t+1)}$ in (5) we have

$$
\begin{aligned}
s_{\pi^{(t+1)}(i)}^{(t+1)} - g_i^{(t+1)} &= s_{\pi^{(t+1)}(i)}^{(t)} - g_i^{(t)} + \eta\gamma\nabla_g l_{\text{latent}}(s_{\pi^{(t+1)}(i)}^{(t)}, g_i^{(t)}) \\
&= s_{\pi^{(t+1)}(i)}^{(t)} - g_i^{(t)} - \eta\gamma\left(s_{\pi^{(t+1)}(i)}^{(t)} - g_i^{(t)}\right), \text{ because } l_{\text{latent}} \text{ is the squared Euclidean distance} \\
&= (1 - \eta\gamma)\left(s_{\pi^{(t+1)}(i)}^{(t)} - g_i^{(t)}\right)
\end{aligned}
$$

and so

$$
\begin{aligned}
\sum_i \|s_{\pi^{(t+1)}(i)}^{(t+1)} - g_i^{(t+1)}\|_2 &= (1 - \eta\gamma)\sum_i \|s_{\pi^{(t+1)}(i)}^{(t)} - g_i^{(t)}\|_2 \\
&\leq (1 - \eta\gamma)\sum_i \|s_{\pi^{(t)}(i)}^{(t)} - g_i^{(t)}\|_2, \text{ by the definition of } \pi^{(t+1)} \text{ in (4)}^{.}
\end{aligned}
$$

The desired result then follows through induction. $\qquad\square$

An important implication of the above behavior is that the distance $\|s_{\pi^{(t+1)}(i)}^{(t)} - s_{\pi^{(t)}(i)}^{(t)}\|_2$ due to a **switch** also decays exponentially.

**Theorem E.2.**

$$\|s_{\pi^{(t+1)}(i)}^{(t)} - s_{\pi^{(t)}(i)}^{(t)}\|_2 \leq 2(1 - \eta\gamma)^t \sum_i \|s_{\pi^{(0)}(i)}^{(0)} - g^{(0)}\|_2$$

*Proof.* By triangle inequality, we have

$$
\begin{aligned}
\|s_{\pi^{(t+1)}(i)}^{(t)} - s_{\pi^{(t)}(i)}^{(t)}\|_2 &\leq \|s_{\pi^{(t+1)}(i)}^{(t)} - g_i^{(t)}\|_2 + \|s_{\pi^{(t)}(i)}^{(t)} - g_i^{(t)}\|_2 \\
&\leq \sum_i \|s_{\pi^{(t+1)}(i)}^{(t)} - g_i^{(t)}\|_2 + \sum_i \|s_{\pi^{(t)}(i)}^{(t)} - g_i^{(t)}\|_2 \\
&\leq 2\sum_i \|s_{\pi^{(t)}(i)}^{(t)} - g_i^{(t)}\|_2, \text{ by the definition of } \pi^{(t+1)} \text{ in (4)}
\end{aligned}
$$

and the desired result follows immediately from Lemma E.1. $\qquad\square$

### E.3 Convergence of the main task's loss function

For convenience, we introduce the following notations:

$$\alpha = 1 - \eta\gamma,$$

$$\mathcal{B} = L\eta\left(1 - \frac{L\eta}{2}\right),$$

$$\mathcal{C} = 4L\left(\frac{1}{2} + \frac{3}{\mathcal{B}} + \frac{9L}{2\mu\mathcal{B}^2}\right)\left(\sum_i \|s^{(0)}_{\pi^{(0)}(i)} - g^{(0)}\|_2\right)^2,$$

$$\delta = 1 - \frac{\mu}{9L}\left(2\mathcal{B}^3 - 13\mathcal{B}^2 + 12\mathcal{B}\right),$$

$$d_t = \|s^{(t)}_{\pi^{(t+1)}(i)} - s^{(t)}_{\pi^{(t)}(i)}\|_2,$$

$$x_t = l_{\text{task}}(s^{(t)}_{\pi^{(t)}(i)}, y_i) - l_{\text{task}}(s^*_i, y_i),$$

where $\alpha$, $\mathcal{B}$, $\mathcal{C}$, and $\delta$ are constants and $d_t$ and $x_t$ are sequences of positive real numbers. If $\eta$ is chosen to be smaller than $\min\{1/\gamma, 2/L\}$, then $\alpha \in (0,1)$ and $\mathcal{B}, \mathcal{C} > 0$. With AM-GM inequality, it is clear that $\mathcal{B} \leq 1/2$. In fact, $\mathcal{B}$ can be made arbitrarily small from the choice of $\eta$. Furthermore, since the cubic polynomial $p(x) = 2x^3 - 13x^2 + 12x$ is increasing on $[0, 1/2]$, we can set $\eta$ to make $\mathcal{B}$ small enough that $p(\mathcal{B}) < 9L/\mu$ and ensure that $\delta \in (0,1)$.

In the proofs below, we rely on the properties of these constants, especially that $\alpha, \delta \in (0,1)$, to show the convergence of LSP.

**Lemma E.3.** *If $l_{task}(\cdot, y)$ is $L$-smooth, we have the following inequalities*

$$\|\nabla_s l_{task}(s^{(t)}_{\pi^{(t+1)}(i)}, y_i)\|_2 \geq \|\nabla_s l_{task}(s^{(t)}_{\pi^{(t)}(i)}, y_i)\|_2 - Ld_t, \tag{9}$$

$$l_{task}(s^{(t+1)}_{\pi^{(t+1)}(i)}, y_i) - l_{task}(s^{(t)}_{\pi^{(t)}(i)}, y_i) \leq \frac{L}{2}d_t^2 + \|\nabla_s l_{task}(s^{(t)}_{\pi^{(t)}(i)}, y_i)\|_2 \cdot d_t - \frac{\mathcal{B}}{L}\|\nabla_s l_{task}(s^{(t)}_{\pi^{(t+1)}(i)}, y_i)\|_2^2. \tag{10}$$

*Proof.* From L-smoothness (6), we have

$$L \cdot \|s^{(t)}_{\pi^{(t+1)}(i)} - s^{(t)}_{\pi^{(t)}(i)}\|_2 \geq \|\nabla_s l_{\text{task}}(s^{(t)}_{\pi^{(t+1)}(i)}, y_i) - \nabla_s l_{\text{task}}(s^{(t)}_{\pi^{(t)}(i)}, y_i)\|_2$$

$$\geq \|\nabla_s l_{\text{task}}(s^{(t)}_{\pi^{(t)}(i)}, y_i)\|_2 - \|\nabla_s l_{\text{task}}(s^{(t)}_{\pi^{(t+1)}(i)}, y_i)\|_2$$

by the triangle inequality. This proves the first inequality.

For the second inequality, we apply L-smoothness (7) with $x = s^{(t)}_{\pi^{(t+1)}(i)}$ and $y = s^{(t+1)}_{\pi^{(t+1)}(i)}$ and obtain

$$l_{\text{task}}(s^{(t+1)}_{\pi^{(t+1)}(i)}, y_i) - l_{\text{task}}(s^{(t)}_{\pi^{(t+1)}(i)}, y_i) \leq \langle \nabla_s l_{\text{task}}(s^{(t)}_{\pi^{(t+1)}(i)}, y_i), s^{(t+1)}_{\pi^{(t+1)}(i)} - s^{(t)}_{\pi^{(t+1)}(i)}\rangle + \frac{L}{2} \cdot \|s^{(t+1)}_{\pi^{(t+1)}(i)} - s^{(t)}_{\pi^{(t+1)}(i)}\|_2^2$$

$$= \langle \nabla_s l_{\text{task}}(s^{(t)}_{\pi^{(t+1)}(i)}, y_i), -\eta\nabla_s l_{\text{task}}(s^{(t)}_{\pi^{(t+1)}(i)}, y_i)\rangle + \frac{L}{2}\|\eta\nabla_s l_{\text{task}}(s^{(t)}_{\pi^{(t+1)}(i)}, y_i)\|_2^2$$

$$= -\eta\left(1 - \frac{L\eta}{2}\right)\|\nabla_s l_{\text{task}}(s^{(t)}_{\pi^{(t+1)}(i)}, y_i)\|_2^2$$

$$= -\frac{\mathcal{B}}{L}\|\nabla_s l_{\text{task}}(s^{(t)}_{\pi^{(t+1)}(i)}, y_i)\|_2^2.$$

We also consider (7) when $x = s^{(t)}_{\pi^{(t)}(i)}$ and $y = s^{(t)}_{\pi^{(t+1)}(i)}$

$$l_{\text{task}}(s^{(t)}_{\pi^{(t+1)}(i)}, y_i) - l_{\text{task}}(s^{(t)}_{\pi^{(t)}(i)}, y_i) \leq \langle \nabla_s l_{\text{task}}(s^{(t)}_{\pi^{(t)}(i)}, y_i), s^{(t)}_{\pi^{(t+1)}(i)} - s^{(t)}_{\pi^{(t)}(i)}\rangle + \frac{L}{2} \cdot \|s^{(t)}_{\pi^{(t+1)}(i)} - s^{(t)}_{\pi^{(t)}(i)}\|_2^2$$

$$\leq \|\nabla_s l_{\text{task}}(s^{(t)}_{\pi^{(t)}(i)}, y_i)\|_2 \cdot d_t + \frac{L}{2}d_t^2,$$

where we use Cauchy-Schwarz inequality on the first term. Adding the two inequalities together gives us the desired result.

$\square$

We are now ready to prove the main Theorem 4.1. With the above notation, we rewrite the inequality as

$$x_{t+1} = l_{\text{task}}(s^{(t+1)}_{\pi^{(t+1)}(i)}, y_i) - l_{\text{task}}(s^*_i, y_i) \leq \begin{cases} \mathcal{C}\,\alpha^{2t}, & \text{if } \|\nabla_s l_{\text{task}}(s^{(t)}_{\pi^{(t)}(i)}, y_i)\| \leq \frac{3Ld_t}{\mathcal{B}} \\ \delta\,x_t, & \text{otherwise} \end{cases}$$

*Proof of Theorem 4.1.* In the first case, we assume $\|\nabla_s l_{\text{task}}(s^{(t)}_{\pi^{(t)}(i)}, y_i)\|_2 \leq \frac{3Ld_t}{\mathcal{B}}$. From the Polyak-Lojasiewicz condition (8), we have

$$l_{\text{task}}(s^{(t)}_{\pi^{(t)}(i)}, y_i) - l_{\text{task}}(s^*_i, y_i) \leq \frac{1}{2\mu}\|\nabla_s l_{\text{task}}(s^{(t)}_{\pi^{(t)}(i)}, y_i)\|_2^2 \leq \frac{9L^2}{2\mu\mathcal{B}^2}d_t^2.$$

From (10) in Lemma E.3, we also have

$$l_{\text{task}}(s^{(t+1)}_{\pi^{(t+1)}(i)}, y_i) - l_{\text{task}}(s^{(t)}_{\pi^{(t)}(i)}, y_i) \leq \frac{L}{2}d_t^2 + \|\nabla_s l_{\text{task}}(s^{(t)}_{\pi^{(t)}(i)}, y_i)\|_2 \cdot d_t - \frac{\mathcal{B}}{L}\|\nabla_s l_{\text{task}}(s^{(t)}_{\pi^{(t+1)}(i)}, y_i)\|_2^2$$

$$\leq \frac{L}{2}d_t^2 + \|\nabla_s l_{\text{task}}(s^{(t)}_{\pi^{(t)}(i)}, y_i)\|_2 \cdot d_t$$

$$\leq \frac{L}{2}d_t^2 + \frac{3L}{\mathcal{B}}d_t^2.$$

We combine above inequalities and Theorem E.2 to conclude that

$$x_{t+1} \leq L\left(\frac{1}{2} + \frac{3}{\mathcal{B}} + \frac{9L}{2\mu\mathcal{B}^2}\right)d_t^{\,2}$$

$$\leq L\left(\frac{1}{2} + \frac{3}{\mathcal{B}} + \frac{9L}{2\mu\mathcal{B}^2}\right)\left(2(1-\eta\gamma)^t \sum_i \|s^{(0)}_{\pi^{(0)}(i)} - g^{(0)}\|_2\right)^2 = \mathcal{C}\,\alpha^{2t}$$

In the second case, we assume $\|\nabla_s l_{\text{task}}(s^{(t)}_{\pi^{(t)}(i)}, y_i)\|_2 > \frac{3Ld_t}{\mathcal{B}}$. Notice that $\mathcal{B}$ is chosen to be small and in particular $1/\mathcal{B} > 2$. From (9) in Lemma E.3, this implies

$$\|\nabla_s l_{\text{task}}(s^{(t)}_{\pi^{(t+1)}(i)}, y_i)\|_2 \geq \|\nabla_s l_{\text{task}}(s^{(t)}_{\pi^{(t)}(i)}, y_i)\|_2 - Ld_t > 5Ld_t \geq 0.$$

We now consider (10) from Lemma E.3

$$l_{\text{task}}(s^{(t+1)}_{\pi^{(t+1)}(i)}, y_i) - l_{\text{task}}(s^{(t)}_{\pi^{(t)}(i)}, y_i) \leq \frac{L}{2}d_t^2 + \|\nabla_s l_{\text{task}}(s^{(t)}_{\pi^{(t)}(i)}, y_i)\|_2 \cdot d_t - \frac{\mathcal{B}}{L}\|\nabla_s l_{\text{task}}(s^{(t)}_{\pi^{(t+1)}(i)}, y_i)\|_2^2$$

$$\leq \frac{L}{2}d_t^2 + \|\nabla_s l_{\text{task}}(s^{(t)}_{\pi^{(t)}(i)}, y_i)\|_2 \cdot d_t - \frac{\mathcal{B}}{L}\left(\|\nabla_s l_{\text{task}}(s^{(t)}_{\pi^{(t)}(i)}, y_i)\|_2 - Ld_t\right)^2$$

$$= \left(\frac{L}{2} - \mathcal{B}L\right)d_t^2 + (1 + 2\mathcal{B})\|\nabla_s l_{\text{task}}(s^{(t)}_{\pi^{(t)}(i)}, y_i)\|_2 \cdot d_t - \frac{\mathcal{B}}{L}\|\nabla_s l_{\text{task}}(s^{(t)}_{\pi^{(t)}(i)}, y_i)\|_2^2.$$

Next, we use our assumption that $B < 1/2$ and $d_t < \frac{\mathcal{B}}{3L}\|\nabla_s l_{\text{task}}(s^{(t)}_{\pi^{(t)}(i)}, y_i)\|_2$ to obtain

$$l_{\text{task}}(s^{(t+1)}_{\pi^{(t+1)}(i)}, y_i) - l_{\text{task}}(s^{(t)}_{\pi^{(t)}(i)}, y_i) < \left(\left(\frac{L}{2} - \mathcal{B}L\right)\frac{\mathcal{B}^2}{9L^2} + (1 + 2\mathcal{B})\frac{\mathcal{B}}{3L} - \frac{\mathcal{B}}{L}\right)\|\nabla_s l_{\text{task}}(s^{(t)}_{\pi^{(t)}(i)}, y_i)\|_2^2$$

$$= -\frac{1}{18L}\left(2\mathcal{B}^3 - 13\mathcal{B}^2 + 12\mathcal{B}\right)\|\nabla_s l_{\text{task}}(s^{(t)}_{\pi^{(t)}(i)}, y_i)\|_2^2$$

$$\leq -\frac{\mu}{9L}\left(2\mathcal{B}^3 - 13\mathcal{B}^2 + 12\mathcal{B}\right)\left(l_{\text{task}}(s^{(t)}_{\pi^{(t)}(i)}, y_i) - l_{\text{task}}(s^*_i, y_i)\right)$$

$$= (\delta - 1)\left(l_{\text{task}}(s^{(t)}_{\pi^{(t)}(i)}, y_i) - l_{\text{task}}(s^*_i, y_i)\right)$$

where we use the Polyak-Lojasiewicz condition (8) and the fact that $p(x) = 2x^3 - 13x^2 + 12x$ is positive on $(0, 1/2]$. Hence $x_{t+1} - x_t < (\delta - 1)x_t$ or $x_{t+1} < \delta\,x_t$ as desired.

$\square$

**Theorem E.4.** *(Convergence of latent set prediction) If the main task's loss function, $l_{task}(\cdot, y)$ is **L-smooth** and satisfies the **Polyak-Lojasiewicz** condition, and $\eta$ is sufficiently small, then the training dynamics described in (5) guarantees its convergence at a linear rate*

$$x_t = l_{task}(s^{(t)}_{\pi^{(t)}(i)}, y_i) - l_{task}(s^*_i, y_i) \leq \mathcal{C}_0 \epsilon^{t-1},$$

*where $\mathcal{C}_0 := \max\{\mathcal{C}, \delta x_0\}$ and $\epsilon := \max\{\alpha^2, \delta\}$.*

*Proof.* This is a consequence of Theorem 4.1. It is easy to check when $t = 1$ and we will proceed by induction.

Suppose that $x_t \leq \mathcal{C}_0 \epsilon^{t-1}$ and consider two cases. If $\|\nabla_s l_{task}(s^{(t)}_{\pi^{(t)}(i)}, y_i)\| \leq \frac{3Ld_t}{\mathcal{B}}$, we have

$$x_{t+1} \leq \mathcal{C}\, \alpha^{2t} \leq \mathcal{C}_0 \epsilon^t.$$

If $\|\nabla_s l_{task}(s^{(t)}_{\pi^{(t)}(i)}, y_i)\| > \frac{3Ld_t}{\mathcal{B}}$, we have

$$x_{t+1} \leq \delta\, x_t \leq \mathcal{C}_0 \delta \epsilon^{t-1} \leq \mathcal{C}_0 \epsilon^t.$$

Therefore $x_t$ is bounded above by an exponential decay. This implies that the main task's loss function converges at least at a linear rate $\epsilon < 1$. This finishes the proof. $\qquad\square$

### E.4 Impact of $\beta > 0$

Setting $\beta > 0$ adds a new term to the update of $s^{(t+1)}_{\pi^{(t+1)}(i)}$ in (5)

$$\begin{aligned}
s^{(t+1)}_{\pi^{(t+1)}(i)} &= s^{(t)}_{\pi^{(t+1)}(i)} - \eta\left(\nabla_s l_{task}(s^{(t)}_{\pi^{(t+1)}(i)}, y_i) + \beta \nabla_s l_{latent}(s^{(t)}_{\pi^{(t+1)}(i)}, g^{(t)}_i)\right) \\
g^{(t+1)}_i &= g^{(t)}_i - \eta\left(\nabla_s l_{task}(s^{(t)}_{\pi^{(t+1)}(i)}, y_i) + \gamma \nabla_g l_{latent}(s^{(t)}_{\pi^{(t+1)}(i)}, g^{(t)}_i)\right).
\end{aligned} \tag{11}$$

which only slightly change the algebra inside the proof of Lemma E.1 as follows

$$\begin{aligned}
s^{(t+1)}_{\pi^{(t+1)}(i)} - g^{(t+1)}_i &= s^{(t)}_{\pi^{(t+1)}(i)} - \eta\beta \nabla_s l_{latent}(s^{(t)}_{\pi^{(t+1)}(i)}, g^{(t)}_i) - g^{(t)}_i + \eta\gamma \nabla_g l_{latent}(s^{(t)}_{\pi^{(t+1)}(i)}, g^{(t)}_i) \\
&= s^{(t)}_{\pi^{(t+1)}(i)} - g^{(t)}_i - \eta(\beta + \gamma)\left(s^{(t)}_{\pi^{(t+1)}(i)} - g^{(t)}_i\right) \\
&= (1 - \eta(\beta + \gamma))\left(s^{(t)}_{\pi^{(t+1)}(i)} - g^{(t)}_i\right)
\end{aligned}$$

Hence, we can derive stronger exponential distance decays in Lemma E.1 and Theorem E.2

**Lemma E.5.** *For $0 < \eta\gamma \leq 1$,*

$$\sum_i \|s^{(t)}_{\pi^{(t)}(i)} - g^{(t)}_i\|_2 \leq (1 - \eta(\beta + \gamma))^t \sum_i \|s^{(0)}_{\pi^{(0)}(i)} - g^{(0)}\|_2.$$

**Theorem E.6.**

$$d_t = \|s^{(t)}_{\pi^{(t+1)}(i)} - s^{(t)}_{\pi^{(t)}(i)}\|_2 \leq 2(1 - \eta(\beta + \gamma))^t \sum_i \|s^{(0)}_{\pi^{(0)}(i)} - g^{(0)}\|_2$$

Next, we note that this also introduces an additional term to the Equation (10) in Lemma E.3 because the derivation of Equation (10) contains the dot product $\langle \nabla_s l_{task}(s^{(t)}_{\pi^{(t+1)}(i)}, y_i), s^{(t+1)}_{\pi^{(t+1)}(i)} - s^{(t)}_{\pi^{(t+1)}(i)} \rangle$. This dot product now contains an additional term $-\eta\beta\langle \nabla_s l_{task}(s^{(t)}_{\pi^{(t+1)}(i)}, y_i), \nabla_s l_{latent}(s^{(t)}_{\pi^{(t+1)}(i)}, g^{(t)}_i) \rangle$ which can be bounded from above by $\eta\beta \left\|\nabla_s l_{task}(s^{(t)}_{\pi^{(t+1)}(i)}, y_i)\right\|_2 \cdot \left\|s^{(t)}_{\pi^{(t+1)}(i)} - g^{(t)}_i\right\|_2$.

Since the rest of the algebra in the proof of Lemma E.3 remains the same, we can readily revise the result of Equation (10) as follows

**Lemma E.7.** *If $l_{task}(\cdot, y)$ is L-smooth, we have the following inequalities*

$$l_{task}(s^{(t+1)}_{\pi^{(t+1)}(i)}, y_i) - l_{task}(s^{(t)}_{\pi^{(t)}(i)}, y_i) \le \frac{L}{2}d_t^2 + \|\nabla_s l_{task}(s^{(t)}_{\pi^{(t)}(i)}, y_i)\|_2 \cdot d_t - \frac{\mathcal{B}}{L}\|\nabla_s l_{task}(s^{(t)}_{\pi^{(t+1)}(i)}, y_i)\|_2^2$$
$$+ \eta\beta \left\|\nabla_s l_{task}(s^{(t)}_{\pi^{(t+1)}(i)}, y_i)\right\|_2 \cdot \left\|s^{(t)}_{\pi^{(t+1)}(i)} - g_i^{(t)}\right\|_2.$$

As an auxiliary result, it is clear that

$$
\begin{aligned}
\left\|s^{(t)}_{\pi^{(t+1)}(i)} - g_i^{(t)}\right\|_2 &\le \sum_i \|s^{(t)}_{\pi^{(t+1)}(i)} - g_i^{(t)}\|_2 \\
&\le \sum_i \|s^{(t)}_{\pi^{(t)}(i)} - g_i^{(t)}\|_2, \text{ by the definition of } \pi^{(t+1)} \text{ in (4)} \qquad (12) \\
&\le (1 - \eta(\beta + \gamma))^t \sum_i \|s^{(0)}_{\pi^{(0)}(i)} - g^{(0)}\|_2, \text{ by Lemma E.5}
\end{aligned}
$$

For convenience, we also introduce some new notations

$$\omega = 1 - \eta(\beta + \gamma),$$
$$\mathcal{D} = \sum_i \|s^{(0)}_{\pi^{(0)}(i)} - g^{(0)}\|_2,$$

where $\omega \in (0,1)$ given appropriate choices of $\beta$, $\eta$, and $\gamma$. This also implies that $\eta\beta \in (0,1)$.

Now, we are ready to revise the proof of the main Theorem 4.1 to incorporate the effect of $\beta > 0$. First, we note that the second case of the proof where $\left\|\nabla_s l_{task}(s^{(t)}_{\pi^{(t)}(i)}, y_i)\right\|_2 > \frac{3Ld_t}{B}$ is when the new $\left\|s^{(t)}_{\pi^{(t+1)}(i)} - g_i^{(t)}\right\|_2$ term interferes with our ability to manipulate the algebra to apply the **Polyak-Lojasiewicz** condition. This is mainly because the condition $\left\|\nabla_s l_{task}(s^{(t)}_{\pi^{(t)}(i)}, y_i)\right\|_2 > \frac{3Ld_t}{B}$ does not tell us much about the value of $\left\|s^{(t)}_{\pi^{(t+1)}(i)} - g_i^{(t)}\right\|_2$. However, as both $d_t$ and $\left\|s^{(t)}_{\pi^{(t+1)}(i)} - g_i^{(t)}\right\|_2$ are bounded above by constant factors of $\omega^t$ (Theorem E.6), we can address this issue by changing the threshold on $\left\|\nabla_s l_{task}(s^{(t)}_{\pi^{(t)}(i)}, y_i)\right\|_2$ for dividing the proof into two cases from $\frac{3Ld_t}{B}$, which uses $d_t$ as reference, to $\frac{6L\mathcal{D}}{\mathcal{B}}\omega^t$, which uses $\omega^t$ as reference.

For the first case where $\left\|\nabla_s l_{task}(s^{(t)}_{\pi^{(t)}(i)}, y_i)\right\|_2 \le \frac{6L\mathcal{D}}{\mathcal{B}}\omega^t$, we have

$$l_{task}(s^{(t)}_{\pi^{(t)}(i)}, y_i) - l_{task}(s_i^*, y_i) \le \frac{1}{2\mu}\|\nabla_s l_{task}(s^{(t)}_{\pi^{(t)}(i)}, y_i)\|_2^2 \le \frac{36L^2\mathcal{D}^2}{\mathcal{B}^2}\omega^{2t}.$$

from the **Polyak-Lojasiewicz** condition, and

$$
\begin{aligned}
l_{task}(s^{(t+1)}_{\pi^{(t+1)}(i)}, y_i) - l_{task}(s^{(t)}_{\pi^{(t)}(i)}, y_i) &\le \frac{L}{2}d_t^2 + \frac{6L\mathcal{D}}{\mathcal{B}}\omega^t d_t + \eta\beta\left\|\nabla_s l_{task}(s^{(t)}_{\pi^{(t+1)}(i)}, y_i)\right\|_2 \cdot \left\|s^{(t)}_{\pi^{(t+1)}(i)} - g_i^{(t)}\right\|_2 \\
&\le \frac{L}{2}d_t^2 + \frac{6L\mathcal{D}}{\mathcal{B}}\omega^t d_t + \frac{6L\mathcal{D}\eta\beta}{B}\omega^t\left\|s^{(t)}_{\pi^{(t+1)}(i)} - g_i^{(t)}\right\|_2.
\end{aligned}
$$

from Lemma E.7. Since both $d_t$ and $\left\|s^{(t)}_{\pi^{(t+1)}(i)} - g_i^{(t)}\right\|_2$ are bounded above by constant factors $\omega^t$ (Theorem E.6), all terms in the above inequalities are bounded above by constant factors of $\omega^{2t}$ and the desired result follows.

For the second case where $\left\|\nabla_s l_{task}(s^{(t)}_{\pi^{(t)}(i)}, y_i)\right\|_2 > \frac{6L\mathcal{D}}{\mathcal{B}}\omega^t$, by applying Theorem E.6, we can show that $\left\|\nabla_s l_{task}(s^{(t)}_{\pi^{(t)}(i)}, y_i)\right\|_2 > \frac{3Ld_t}{\mathcal{B}}$ as in the original proof. Hence, we can follow the same algebraic manipulations to derive

$$
\begin{aligned}
l_{task}(s^{(t+1)}_{\pi^{(t+1)}(i)}, y_i) - l_{task}(s^{(t)}_{\pi^{(t)}(i)}, y_i) &< -\frac{1}{18L}\left(2\mathcal{B}^3 - 13\mathcal{B}^2 + 12\mathcal{B}\right)\|\nabla_s l_{task}(s^{(t)}_{\pi^{(t)}(i)}, y_i)\|_2^2 \\
&+ \eta\beta\left\|\nabla_s l_{task}(s^{(t)}_{\pi^{(t+1)}(i)}, y_i)\right\|_2 \cdot \left\|s^{(t)}_{\pi^{(t+1)}(i)} - g_i^{(t)}\right\|_2
\end{aligned}
$$

From (12) and the condition of this second case, we can bound the last term from above by the gradient of task loss

$$\left\| s^{(t)}_{\pi^{(t+1)}(i)} - g^{(t)}_i \right\|_2 \leq \mathcal{D}\omega^t < \frac{\mathcal{B}}{6L} \left\| \nabla_s l_{\text{task}}(s^{(t)}_{\pi^{(t)}(i)}, y_i) \right\|_2$$

This yields

$$l_{\text{task}}(s^{(t+1)}_{\pi^{(t+1)}(i)}, y_i) - l_{\text{task}}(s^{(t)}_{\pi^{(t)}(i)}, y_i) < -\frac{1}{18L} \left(2\mathcal{B}^3 - 13\mathcal{B}^2 + 12\mathcal{B} - 3\eta\beta\mathcal{B}\right) \|\nabla_s l_{\text{task}}(s^{(t)}_{\pi^{(t)}(i)}, y_i)\|_2^2$$

Because $\eta\beta \in (0,1)$, we can show that for $x \in (0,1/2]$, the polynomial $p(x) = 2x^3 - 13x^2 + 12x - 3\eta\beta x$ is strictly greater than $q(x) = 2x^3 - 13x^2 + 9x$ which is always positive in this interval. Hence, we can follow the original proof to get the desired result.

## E.5 Impact of gradient cloning with rejection

The rejection mechanism adds complexity to the variable update equation (5) by modifying the task loss gradient with

$$\nabla_{s_\pi} \mathcal{L}_{\text{task, rejected}} = \nabla_{s_\pi} \mathcal{L}_{\text{task}} - \frac{\nabla_{s_\pi} \mathcal{L}_{\text{latent}} \cdot \langle \nabla_{s_\pi} \mathcal{L}_{\text{task}}, \nabla_{s_\pi} \mathcal{L}_{\text{latent}}\rangle}{\|\nabla_{s_\pi} \mathcal{L}_{\text{latent}}\|_2^2}$$

on either the update of $s^{(t+1)}_{\pi^{(t+1)}(i)}$ or $g^{(t+1)}_i$ depending on which variable is leading (Figure 3).

Since gradient rejection is activated when $\|\nabla_{s_\pi} \mathcal{L}_{\text{latent}} - \nabla_g \mathcal{L}_{\text{latent}}\|_2 > d \|\nabla_{s_\pi} \mathcal{L}_{\text{task}}\|_2$, we can derive an upper bound for this modification term as follows

$$
\begin{aligned}
\left\| \frac{\nabla_{s_\pi} \mathcal{L}_{\text{latent}} \cdot \langle \nabla_{s_\pi} \mathcal{L}_{\text{task}}, \nabla_{s_\pi} \mathcal{L}_{\text{latent}}\rangle}{\|\nabla_{s_\pi} \mathcal{L}_{\text{latent}}\|_2^2} \right\|_2 &\leq \|\nabla_{s_\pi} \mathcal{L}_{\text{task}}\|_2 \\
&< \frac{1}{d} \|\nabla_{s_\pi} \mathcal{L}_{\text{latent}} - \nabla_g \mathcal{L}_{\text{latent}}\|_2 \\
&= \frac{1}{d}(\beta + \gamma) \left\| s^{(t)}_{\pi^{(t+1)}(i)} - g^{(t)}_i \right\|_2
\end{aligned}
\tag{13}
$$

The proof of Lemma E.5 can be modified to incorporate this term by

$$
\begin{aligned}
\left\| s^{(t+1)}_{\pi^{(t+1)}(i)} - g^{(t+1)}_i \right\|_2 &\leq \omega \left\| s^{(t)}_{\pi^{(t+1)}(i)} - g^{(t)}_i \right\|_2 + \left\| \frac{\nabla_{s_\pi} \mathcal{L}_{\text{latent}} \cdot \langle \nabla_{s_\pi} \mathcal{L}_{\text{task}}, \nabla_{s_\pi} \mathcal{L}_{\text{latent}}\rangle}{\|\nabla_{s_\pi} \mathcal{L}_{\text{latent}}\|_2^2} \right\|_2 \\
&\leq \left(\omega + \frac{1}{d}(\beta + \gamma)\right) \left\| s^{(t)}_{\pi^{(t+1)}(i)} - g^{(t)}_i \right\|_2
\end{aligned}
$$

where $d$ must satisfy the conditions $d > 1/\eta$ to ensure that the constant factor lies in $(0,1)$.

Similarly to the impact of setting $\beta > 0$, the modification of the task loss gradient can appear as an extra term in Lemma E.7 through the dot product $\langle \nabla_s l_{\text{task}}(s^{(t)}_{\pi^{(t+1)}(i)}, y_i), s^{(t+1)}_{\pi^{(t+1)}(i)} - s^{(t)}_{\pi^{(t+1)}(i)}\rangle$ if the rejection mechanism is activated and $s^{(t)}_{\pi^{(t+1)}(i)}$ is leading. Interestingly, this extra term is easy to handle as

$$\langle \nabla_s l_{\text{task}}(s^{(t)}_{\pi^{(t+1)}(i)}, y_i), -\frac{\nabla_{s_\pi} \mathcal{L}_{\text{latent}} \cdot \langle \nabla_{s_\pi} \mathcal{L}_{\text{task}}, \nabla_{s_\pi} \mathcal{L}_{\text{latent}}\rangle}{\|\nabla_{s_\pi} \mathcal{L}_{\text{latent}}\|_2^2}\rangle = -\frac{\langle \nabla_{s_\pi} \mathcal{L}_{\text{task}}, \nabla_{s_\pi} \mathcal{L}_{\text{latent}}\rangle^2}{\|\nabla_{s_\pi} \mathcal{L}_{\text{latent}}\|_2^2}$$

which is strictly negative. Hence, even if the rejection mechanism is activated, the result of the Lemma E.7 still holds in its current form. This implies that the proof of the main theorem also holds.

## F Computational resources

- A typical training time of models on the MNIST point cloud autoencoding task was around 2.5 GPU hours on an RTX 2080Ti.

- A typical training time of models on the CLEVR object description generation task was around 2 GPU hours on a V100.
- A typical training time of models on the chest radiograph report generation task was around 5 GPU hours on an A100.
- A typical training time of models on the object detection task was around 8 GPU hours on an RTX 3090 (batch size 8).