# OpenReview forum: "Set Prediction in the Latent Space"
_NeurIPS.cc/2021/Conference — NeurIPS 2021 Poster_

### Official Review · Reviewer_U4qy · 2021-07-14

**Rating:** 6
**Confidence:** 5

**Summary:**

Previous set prediction networks have typically been learned by using a matching step (e.g. Hungarian algorithm, chamfer distance) post-output in order to match the predicted set with the target set before computing the loss and propagating the gradient. Because the sets are represented as vectors (one per set element) stacked up in matrices, the predicted and target sets have an arbitrary order that might not be compatible with each other to directly compare corresponding elements (that is there might be a permutation that must be applied to one of the sets in order to align the elements with the other).

This paper proposes to move the matching step inside the network during training, such that a learned representation of each target element (in the order provided by the target) is produced that can be matched with the latent representation of each predicted set element. As such, when the latent vectors corresponding to the input are decoded they are in an order that is compatible with the ordering of the target set.   The inherent problem that arises is that there are discontinuities when assignments between the target and predicted sets change, so this work proposes a neat workaround that allows the target encodings to move towards and 'chase' their matched predicted latent.


**Limitations And Societal Impact:**

Discussion is adequate, although further evaluation would help with understanding the limitations.

**Main Review:**

Originality
-----------

Overall, I believe that this is an original take on solving the set prediction problem, and the approach to circumventing the problem of discontinuities is new and intuitive.


Quality
-------

I believe that the key idea proposed in the paper is technically sound, and although I didn't work the idealized convergence proof through in full detail, in sketch form it seems reasonable. The empirical results presented also seem to suggest good convergence with real-world tasks.

I am however very concerned that there appears to be no comparison against the current alternative general set prediction models (e.g. TSPN and DSPN) for tasks that don't require teaching forcing and where the authors have already the specified distance metrics used for the Hungarian matching (negating the need to hand-craft anything). I do not for example see why a comparison against all the experiments in Zhang et al's DSPN paper could not be performed with similar network architectures (ditto TSPN). Such an comparison would not just look at the set-prediction performance, but also compare e.g. training time, convergence, etc.

I also question if the claim about hand-crafted distances for the Hungarian algorithm being _required_ in prior set prediction work and not for the proposed approach, rather overstated? From my understanding none of the existing PIT methods preclude the possibility of using a learned distance function.

Clarity
-------

The overall presentational clarity of the paper could be improved. There are a considerable number of awkwardly written sentences and grammatical errors that make digesting the content significantly harder than it needs to be.  Similar understandability problems manifest themselves in the figures, which could also be improved (together with the captions). For example:

- Figure 1.  Would be good to illustrate exactly what inference looks like compared to training (unless one understands the whole model, the last sentence of the caption only confuses matters)
- Figure 2. Maybe show the y's and drop the gradient labels? Alternatively rewrite the caption to talk in terms of the gradient.

In the introduction (and at a few points throughout the paper), the claim that current set prediction models cannot be used with teacher forcing is made; having read through the paper multiple times I think I now understand - is the argument that you might want to decode a set of sequences, whereby it might be useful to perform teacher forcing whilst learning to decode; however in previous settings you couldn't do this because you needed to fully decode in order to perform the matching? If so, perhaps a concrete example of a task could be given (and added to the introduction)?

 The current experiments section comes across as a hodgepodge of different evaluations. I believe this could be much better structured with a comparative evaluation against alternative techniques, followed by experiments that demonstrate fully the advantages (with respect to e.g. teacher forcing and distance metrics) of the proposed framework over those existing methods.

Significance
------------

I think there is potential for this approach to set prediction to have an impact, however as I've outlined above there is a need to fairly compare it against other alternative general set prediction models so the advantages (and potentially disadvantages can be fully assessed). Without such an evaluation it is not in any way clear if the proposed approach is really only useful for when teacher forcing is required, or hand-crafted distances are to be avoided (caveated on the question above).


Rationale for (original) score
-------------------

I want to make it clear that I completely believe that there are some interesting ideas in this work. However at the same time, in its current form, it is let down by the lack of comparison in the empirical evaluation, and a presentation that is at times hard to follow.

Post rebuttal score change
---------------------

I have raised my score in light of the additional experiments presented in the comments, and the promises to improve clarity.

**Time Spent Reviewing:**

5

---

> ### Author Response · Authors · 2021-08-10
> **Response to the review (Part 1 of 3)**
>
> Thank you for your review. Below is our response to your comments.
>
> **Q1: “I am however very concerned that there appears to be no comparison against the current alternative general set prediction models (e.g. TSPN and DSPN) for tasks that don't require teaching forcing and where the authors have already the specified distance metrics used for the Hungarian matching (negating the need to hand-craft anything). I do not for example see why a comparison against all the experiments in Zhang et al's DSPN paper could not be performed with similar network architectures (ditto TSPN).”
> “there is a need to fairly compare it against other alternative general set prediction models so the advantages (and potentially disadvantages can be fully assessed).”**
>
> We would like to point out that TSPN [1] is similar to DETR [2] which was already included as a baseline in the object detection experiment. Both TSPN and DETR are based on Transformer and DETR was specifically designed for object detection tasks. As TSPN was already shown to outperform DSPN [1, 12] and as DETR (TSPN equivalent) was already included as a baseline, our current results should provide good evidence for the strong performance of LSP.
>
> [1] Adam R Kosiorek, Hyunjik Kim, and Danilo J Rezende, “Conditional set generation with transformers,” June 2020.
>
> [2] Nicolas Carion, FranciscoMassa, Gabriel Synnaeve, et al., “End-to-End object detection with transformers,” May 2020.
>
> [12] Yan Zhang, Jonathon Hare, and Adam Prugel-Bennett, “Deep set prediction networks,” in Advances in Neural Information Processing Systems 32, H Wallach, H Larochelle, A Beygelzimer, et al., Eds., pp. 3212–3222. Curran Associates, Inc., 2019.
>
> ---
> **Q2: “Such an comparison would not just look at the set-prediction performance, but also compare e.g. training time, convergence, etc.”**
>
> We agree that training time and training curves, in particular, may be useful to the readers. We will include these in the final version of the appendix.

---

> ### Author Response · Authors · 2021-08-10
> **Response to the review (Part 2 of 3)**
>
> **Q3: “I also question if the claim about hand-crafted distances for the Hungarian algorithm being required in prior set prediction work and not for the proposed approach, rather overstated? From my understanding none of the existing PIT methods preclude the possibility of using a learned distance function.”**
>
> We confirm that the existing PIT methods do require manually defined distance functions while LSP does not. While it is possible to just use the loss function as the distance function, this is not always the case as evident in DETR [2] using a manually designed distance function.
>
> We also want to emphasize that the choice of distance function is not arbitrary. A poor choice of distance function (either learned on manually designed) can compromise the convergence of set prediction. An ideal distance function gives small values when the loss is small (near convergence) and large values when the loss is large (far from convergence) to strengthen the assignment overtime. Failure to achieve this will result in unnecessary “switches” that slow down training, or in the extreme case, where the distance function does not share minimizers with the loss function, the training does not converge due to perpetual switches. Even when the distance function and the loss function are the same and not learned, the problem of switches has been reported by many works and can be detrimental to the results [R1, R2]. When the distance function is learned, the problem of switches becomes detrimental when not using GC in PIT training as shown in our synthetic dataset experiment.
>
> As you can see, it is not trivial to “learn” a distance function to satisfy the above criteria. Anyone who proposes to learn one must justify his/her choice on the convergence question which we have yet to see any. To the best of our knowledge, LSP is the first approach to learn the distance function that is guaranteed to satisfy the aforementioned criteria in the general domain.
>
> [R1] Huang, Sung-Feng, Shun-Po Chuang, Da-Rong Liu, Yi-Chen Chen, Gene-Ping Yang, and Hung-Yi Lee. 2020. “Stabilizing Label Assignment for Speech Separation by Self-Supervised Pre-Training.” arXiv [cs.SD]. arXiv. http://arxiv.org/abs/2010.15366.
>
> [R2] Yousefi, Midia, Soheil Khorram, and John H. L. Hansen. 2019. “Probabilistic Permutation Invariant Training for Speech Separation.” arXiv [eess.AS]. arXiv. http://arxiv.org/abs/1908.01768.

---

> ### Author Response · Authors · 2021-08-10
> **Response to the review (Part 3 of 3; final)**
>
> **Q4: “I think I now understand - is the argument that you might want to decode a set of sequences, whereby it might be useful to perform teacher forcing whilst learning to decode; however in previous settings you couldn't do this because you needed to fully decode in order to perform the matching? If so, perhaps a concrete example of a task could be given (and added to the introduction)?”**
>
> We agree that this is where the manuscript could be improved regarding the teacher forcing problem explanation.
>
> ---
> **Q5: “The current experiments section comes across as a hodgepodge of different evaluations. I believe this could be much better structured with a comparative evaluation against alternative techniques, followed by experiments that demonstrate fully the advantages (with respect to e.g. teacher forcing and distance metrics) of the proposed framework over those existing methods.”**
>
> We agree with your suggestion to reflow Section 5 (experiments). For the final version of this manuscript, we will start with the “synthetic dataset” which helps the reader understand the characteristics of set prediction in the latent space and LSP, then the “object detection” for comparing LSP with the traditional set prediction pipeline, and lastly the “image captioning” experiments to demonstrate applicability of LSP with teacher forcing which was previously impractical by traditional set prediction.
>
> ---
> **Q6: “It is not in any way clear if the proposed approach is really only useful for when teacher forcing is required, or hand-crafted distances are to be avoided (caveated on the question above).”**
>
> We want to clarify that LSP is useful in at least two scenarios:
>
> 1. Sequence prediction that requires teacher forcing. Before LSP, there are really two ways to tackle this problem: 1) using an ordered set or concat approach which will hurt the performance, 2) using the exhaustive generation which will require $O(N^2)$ generations. After LSP, it is now possible to do set prediction without exhaustive generation requiring only $O(N)$ generations.
>
> 2. The distance function is hard to define while an encoder is easier to define. Before LSP, we need to come up with a distance function. If not exactly, we will use a surrogate or an approximation which may hurt the performance. A learned distance function is possible, yet it raises questions regarding convergence properties which are hard to justify. After LSP, if an encoder is much easier to design, we can take the LSP path without the need to define the distance function at all. LSP lets the main task’s objective function guide the learning of distance function on the fly.

---

> ### Comment · Reviewer_U4qy · 2021-08-27
> **Post rebuttal comment**
>
> First I'd like to thank the authors for taking the time to respond. As it currently stands I am not leaning towards changing my score because I find the response for greater comparison against the other set prediction approaches (made not just by myself, but also other reviewers) rather unsatisfactory.
>
> More specifically, I have problems buying the argument that we should only be considering DETR as a baseline as it's better than TSPN and DSPN for the particular task of object detection. The way the paper is written is to propose LSP as general approach to the set prediction task; if it is truly a general approach, then one would like to understand how it performs on all the tasks that previous approaches have been tested on (or at least a range of them). Object detection isn't the only set prediction task those other works evaluate on, and it also isn't particularly fair to compare vastly different architectures for object detection (for example the DSPN paper used a simple MLP baseline as the set decoder to be comparable with other earlier related work; clearly with a more powerful decoder the performance on object detection would likely be better).

---

> > ### Author Response · Authors · 2021-08-31
> > **Clarifications and additional experiment**
> >
> > We have a few points to clarify and a new experiment to present.
> >
> > **Clarifications**
> >
> > **LSP is indeed general** due to the formulation that does not assume any particular domain of set elements. Yet, LSP requires an **encoder** which can be "not an easy task" (Section 6: Limitation). Because the performance of LSP relies greatly on the encoder, when the distance metric is easy to design and straightforward, the prospect of designing an encoder for LSP is not appealing. Hence, we do not claim that LSP is suitable for all tasks, only the ones that we can design good encoders.
> >
> > **LSP is especially appealing for tasks that were previously hard to do with set prediction.** When a good distance metric is hard to design but the encoder is more straightforward, e.g. set of texts, LSP is an appealing choice breaking through the previous limits of set prediction. We believe that this is the main contribution of LSP as mentioned in the introduction.
> >
> > In short, LSP is applicable to general set prediction regardless of the domain, but LSP is only as good as the encoder. If a good encoder is easy to comeby, LSP is an appealing choice, otherwise, traditional set prediction is the way to go.
> >
> > ---
> > **Point cloud autoencoding experiment**
> >
> > We hope to address your concern by comparing DSPN, TSPN, and LSP under the same dataset.
> > We use the MNIST point cloud dataset following DSPN [2].  We performed comparison only on Hungarian assignment which is different from the DSPN and TSPN papers.  The metric is "sum" Chamfer L2 distance on the test dataset. Experiments were run for 3 seeds. This experiment will be included in the final version of appendix.
> >
> > | **Model** | **Hidden dim** | **Chamfer L2 distance** |
> > | - | - | - |
> > | DSPN [(click to see image)](https://i.ibb.co/2k4rz91/dspn256.png) | 256 | 4.99 ± 0.37 |
> > | DSPN [(click to see image)](https://i.ibb.co/vDHh41F/dspn512.png) | 512| 4.52 ± 0.23 |
> > | DSPN* [(click to see image)](https://i.ibb.co/JkRPpJQ/dspn1024.png) | 1024 | 4.42 ± 0.01 |
> > | TSPN [(click to see image)](https://i.ibb.co/VWtfPKj/tspn.png) | 256 | **3.78** ± 0.01 |
> > | LSP (GCR, d=0.001) [(click to see image)](https://i.ibb.co/rFWwGf5/lsp.png)| 256 | 3.90 ± 0.02 |
> >
> > ± represents one standard deviation. *only 2 seeds.
> >
> > DSPN’s were outperformed by both TSPN and LSP by a large margin which can be observed qualitatively. Also, DSPN did not scale up well with wider models. TSPN performed better than LSP in this experiment, yet qualitatively hard to perceive the differences.
> >
> > We want to point out that this is possibly a task where designing a good encoder is harder than designing a good distance metric since each set element is simply (x, y, presence).
> >
> > In our object detection experiment, each set element is a bit more complex being (x, y, w, h, class). The distance metric is harder to design, hence we observed LSP faired better than in this experiment.
> >
> > **Experimental details**
> >
> > The point cloud dataset has the maximum of 150 points. All experiments were run for 50 epochs. Each performance number was the minimum of the run.
> > The three models share the same architecture (3 Conv + FSPool following DSPN [2]) for their set encoders. We varied the “hidden dimension” of the encoder for DSPN to scale it up for comparable parameter count. DSPN’s were run with 0.01 learning rate (following DSPN [2] and swept from [0.01, 0.001, 0.0001]) while the others run with 0.0001 learning rate.
> > We used the batch normalized of the set encoder due to faster convergence.
> > TSPN and LSP have identical architecture except LSP having an encoder which is simply a linear layer from (x, y, presence) to a 256-sized vector. We have found the original TSPN suboptimal for Hungarian assignment. We used the learned embeddings instead of Gaussian random vectors. Also, we predicted each element’s presence to derive the set cardinality instead of using an MLP to predict the number of set cardinality explicitly.
> > The parameter counts are as follows:
> >
> > | **Model**     | **Hidden dim** | **Parameter count** |
> > | - | - | - |
> > | DSPN | 256 | 139970 |
> > | DSPN | 512 | 404418 |
> > | DSPN | 1024 | 1326530 |
> > | TSPN | 256 | 1498371 |
> > | LSP (GCR, d=0.001) | 256 | 1499395 |

---

> > > ### Comment · Reviewer_U4qy · 2021-08-31
> > > **Thank you**
> > >
> > > These additional results go a good way to addressing some of the concerns raised. I have updated my score.

---

### Official Review · Reviewer_BHaN · 2021-07-16

**Rating:** 5
**Confidence:** 3

**Summary:**

This paper proposes a framework for deep-set prediction, they also give a theoretical model for LSP and show the promising results on benchmark datasets. The proposed Latent set prediction looks new.

**Limitations And Societal Impact:**

- In general, I found it is a little bit hard to read this paper, for example, in Sec 3.2, they design a set of losses, but in the experiment, I can not find the abolition study of those losses. They only show the final model.
- As for Sec. 5.4, they verify their method on modified datasets (MNIST) and compare it with DETR, this makes the experiment not strong. Also, why on MNIST, I think there are many bigger datasets better than MNIST, they should use those datasets to support their method.
- For image captioning, why not use MSCOCO as the target dataset? I think MSCOCO is a more standard dataset for image captioning.
- Since this method focus on latent set prediction, I am wondering if this method can be applied for image-text cross-modal retrieval. If can, it would be better to show the effectiveness of latent set on cross-modality matching.
- During image captioning, which loss was used? and some experiment settings for CLEVR object description generation and Chest radiograph report generation are missing.
- The origination of this paper needs to be improved and some equation indexes are missing. such as, in Line 27, what is bijective ?

**Main Review:**

Originality: The latent set prediction looks interesting. They provide an extensive theoretical analysis of their method and achieved good performance on CLEVER and MIIMIC-CXR datasets.

Quality: The experiment results on CLEVR look good, but still low on MIMIC-CXR dataset. Also, they did not consider some more standard datasets on image captioning and detection.

Clarity: The writing of this paper should be improved, with too many bold fonts in the main paragraph, and also should not use red color for the symbols.

Significance: The alleviate the need for hand-crafted distance metrics looks very promising.

**Time Spent Reviewing:**

3

---

> ### Author Response · Authors · 2021-08-10
> **Response to the review (Part 1 of 3)**
>
> Thank you for your review. Below is our response to your comments.
>
> **Q1: “in Sec 3.2, they design a set of losses, but in the experiment, I can not find the abolition study of those losses. They only show the final model.”**
>
> We would like to clarify that LSP requires only one additional loss, which was defined in Equation 2. This is because LSP performed the assignment in the latent space $\mathbb{R}^c$ that is naturally associated with the Euclidean distance. Hence, unlike other methods that require the selection of a distance metric and loss function for the assignment, LSP does not require such extra decisions to be made. The main parameters of LSP are $\beta$, the weight for the latent loss, and $d$  in GCR.
>
> In our preliminary experiment on the CLEVR task, $\beta$ was quite robust and setting it to 0.1 yielded good performances across datasets. The importance of $d$ in GCR was shown in an ablation study. Please note that $L_\text{latent}$ is crucial to the convergence of LSP. Without it, LSP is not even theoretically guaranteed to converge.
>
> ---
> **Q2: “During image captioning, which loss was used?”**
>
> Equation 2 is the only loss function we proposed, $L_\text{total} = L_\text{latent} + L_\text{task}$. In the case of image captioning, $L_\text{task}$ is a cross-entropy loss between the predicted tokens and the ground truth subword tokens usual for NLP tasks. As of $L_\text{latent}$ (asymmetric latent loss), it is the squared Euclidean distance between $s$ and $g$ controlled by a hyperparameter $\beta$ (set as 0.1 by default). This is everything about the loss function.
>
> However, for LSP to work, it also needs GC or GCR to encourage the convergence of $L_\text{task}$ to its local minima. Without either $L_\text{task}$ or GC (or GCR), the latent set prediction will not converge which is shown by our convergence analysis (Section 4) and the synthetic dataset experiment (Section 5.1).

---

> ### Author Response · Authors · 2021-08-10
> **Response to the review (Part 2 of 3)**
>
> **Q3: “some experiment settings for CLEVR object description generation and Chest radiograph report generation are missing.”**
>
> We believe we have put important experimental details in the appendix which includes all training details, model details, and dataset details. We also provide reproducible codes as a part of supplementary.
>
> ---
> **Q4: “The origination of this paper needs to be improved and some equation indexes are missing. such as, in Line 27, what is bijective ?”**
>
> We want to clarify that “bijection” or one-to-one correspondence (see [Wikipedia](https://en.wikipedia.org/wiki/Bijection)) was used to describe a Hungarian matching where each element needs to be mapped without being left out. We are unsure which equation indexes are missing.
>
> ---
> **Q5: “I am wondering if this method can be applied for image-text cross-modal retrieval. If can, it would be better to show the effectiveness of latent set on cross-modality matching.”**
>
> We are not accustomed to the image-text cross-modal retrieval task. We believe that the reviewer refers to an approach like CLIP by Radford et al. which is based on contrastive learning between <image, text> pairs. We do not see any obvious connection between LSP and contrastive learning, especially because “text” in the <image, text> pairs are unlikely to be a set. Hence, LSP may not be applicable as the task is not set prediction.

---

> ### Author Response · Authors · 2021-08-10
> **Response to the review (Part 3 of 3; final)**
>
> **Q6: “For image captioning, why not use MSCOCO as the target dataset? I think MSCOCO is a more standard dataset for image captioning”**
>
> We looked for set prediction tasks that would highlight the new capabilities gained through LSP. First, the sets to be predicted should not be too small (otherwise non-set prediction would suffice). Second, teacher forcing, which was not possible in prior set prediction approaches, should be applicable. Third, the input should be highly predictive of the output to minimize effects from confounding factors. Lastly, we preferred tasks with real-world applications.
>
> MIMIC-CXR (chest x-ray reported generation) was selected because a medical report can be thought of as a set of sentences each describing an observed abnormality. We prefer MIMIC-CXR over standard datasets like MS-COCO because each MS-COCO’s caption usually contains one to two sentences while medical reports are much longer. Furthermore, MIMIC-CXR is less subjective than MS-COCO because while all radiologists receive standardized training to spot the same abnormalities, general image captioning highly depends on the labellers. Most importantly, medical reports [R4, R5] and automatic chest x-ray assessments [R1, R2, R3] have seen great interest in recent years due to its applications in healthcare [R6].
>
> CLEVR was selected mainly because its ground truths are known without any ambiguity nor confounding factor. Hence, it is perfect for comparing LSP against other methods.
>
> **References:**
>
> [R1] Rajpurkar, Pranav, Jeremy Irvin, Kaylie Zhu, Brandon Yang, Hershel Mehta, Tony Duan, Daisy Ding, et al. 2017. “CheXNet: Radiologist-Level Pneumonia Detection on Chest X-Rays with Deep Learning.” arXiv [cs.CV]. arXiv. https://doi.org/1711.05225.
>
> [R2] Irvin, Jeremy, Pranav Rajpurkar, Michael Ko, Yifan Yu, Silviana Ciurea-Ilcus, Chris Chute, Henrik Marklund, et al. 2019. “CheXpert: A Large Chest Radiograph Dataset with Uncertainty Labels and Expert Comparison.” Proceedings of the AAAI Conference on Artificial Intelligence 33 (July): 590–97.
>
> [R3] Horng, Steven, Ruizhi Liao, Xin Wang, Sandeep Dalal, Polina Golland, and Seth J. Berkowitz. 2021. “Deep Learning to Quantify Pulmonary Edema in Chest Radiographs.” Radiology: Artificial Intelligence 3 (2): e190228.
>
> [R4] Chen, Zhihong, Yan Song, Tsung-Hui Chang, and Xiang Wan. 2020. “Generating Radiology Reports via Memory-Driven Transformer.” arXiv [cs.CL]. arXiv. https://www.aclweb.org/anthology/2020.emnlp-main.112.pdf.
>
> [R5] Liu, Guanxiong, Tzu-Ming Harry Hsu, Matthew McDermott, Willie Boag, Wei-Hung Weng, Peter Szolovits, and Marzyeh Ghassemi. 2019. “Clinically Accurate Chest X-Ray Report Generation.” In Proceedings of the 4th Machine Learning for Healthcare Conference, PMLR, edited by Finale Doshi-Velez, Jim Fackler, Ken Jung, David Kale, Rajesh Ranganath, Byron Wallace, and Jenna Wiens, 106:249–69. Proceedings of Machine Learning Research. Ann Arbor, Michigan: PMLR.
>
> [R6] Sim, Yongsik, Myung Jin Chung, Elmar Kotter, Sehyo Yune, Myeongchan Kim, Synho Do, Kyunghwa Han, et al. 2020. “Deep Convolutional Neural Network-Based Software Improves Radiologist Detection of Malignant Lung Nodules on Chest Radiographs.” Radiology 294 (1): 199–209.
>
> ---
> **Q7: “As for Sec. 5.4, they verify their method on modified datasets (MNIST) and compare it with DETR, this makes the experiment not strong. Also, why on MNIST, I think there are many bigger datasets better than MNIST, they should use those datasets to support their method.”**
>
> We would like to point out that our modified MNIST dataset is not trivial as the name may suggest. As shown in appendix C.1 or [(click to see image)](https://i.ibb.co/yNZJg2q/2564-08-05-18-22-53-lsp-v3-full-pdf-Adobe-Acrobat-Pro-DC.png), a single image can contain as many as 50 digits with a lot of overlap and variations in orientations, brightness, contrast, and sharpness. The prediction scores also indicated that this task is not trivial even for a ResNet-50 DETR model (Table 4).

---

> > ### Comment · Reviewer_BHaN · 2021-08-31
> > **Thanks for the rebuttal**
> >
> > Thank you for your detailed answers to my questions, which addressed most of them! To make this work more robust, there is a dataset on natural images (https://google.github.io/localized-narratives/) that provides rich information than COCO, and I suggest the authors consider this in their experiments.

---

> > > ### Author Response · Authors · 2021-08-31
> > > **Thanks for the suggestion**
> > >
> > > Thank you for the dataset suggestion. We agree that natural image captioning tasks would make a more relatable benchmark. The localized narrative dataset generally has a longer caption than MS-COCO. Looking further, the Stanford Visual Paragraphs dataset also has long paragraph descriptions which can make for interesting use cases of LSP. However, we believe that the x-ray report generation task is a good representative of an image captioning task for the sake of showcasing the potential of LSP on real-world datasets, and would like to consider the Visual Paragraph dataset as potential future work.

---

> ### Comment · Area_Chair_c68Q · 2021-08-28
> **The end of the discussion period is approaching**
>
> Dear Reviewer BHaN,
>
> The authors have provided detailed responses to your comments. Please thoroughly go over the author's response provide as soon as possible, and provide feedback, since we have less than a week to have interactive discussions with the authors. Thank you again for your service as a reviewer for NeurIPS 2021.
>
> Thanks,
> Area Chair

---

> ### Comment · Area_Chair_c68Q · 2021-08-31
> **[Urgent] The end of the discussion period is less than two days away**
>
> Dear Reviewer BHaN,
>
> We have less than two days left until the discussion deadline. Please do go over the responses from the authors and provided feedback, and engage in the discussion with the authors and other reviewers.
>
> Thanks,
> Area Chair

---

### Official Review · Reviewer_6XRy · 2021-07-18

**Rating:** 7
**Confidence:** 4

**Summary:**

This paper proposes a novel learning framework for set predictions. Due to its permutation-invariant nature, learning to predict sets usually requires costly matching between ground-truth sets and generated sets. Most of the previous methods do this in the actual data space where the sets reside; based on a pre-specified distance metric, optimal assignments between ground-truth sets and generated sets are computed via Hungarian algorithm, and then the errors are backpropagated with those assignments. As the authors pointed out, it is often hard to hand-specify such distance metrics that can be informative enough to get training signals, especially for high-dimensional data. This paper proposes to conduct the matching between sets in latent space in order to alleviate this problem; Of course, if possible, matching in latent space would make the problem much easier, but now the problem is that matching in latent space requires matching two moving targets in the latent space (representations of ground-truth sets and representations of generated sets). This actually causes the "switching" problem that can hinder the training. The authors an optimization algorithm where the representations of ground-truths and generated sets are kept close via additional loss function and a special mechanism called gradient cloning (with rejection). The proposed algorithm is shown to guarantee convergence.

**Ethics Review Area:**

["I don’t know"]

**Limitations And Societal Impact:**

The paper properly discusses the limitations. The checklist says the paper discusses societal impact but as far as I can see it is not that apparent.

**Main Review:**

I find the paper is very well-written; The paper tackles an important problem that the literature has been overlooking, and provides a sound solution with a theoretical guarantee. I think the proposed solution can potentially inspire other related methods since we often encounter problems where we have to optimize or match two moving targets and fail to make algorithms converge presumably due to a similar reason as the switching problem in the set prediction context.

The experimental results are extensive and interesting. The chest radiograph report generations are quite impressive in my opinion. It would be better though to include more recent baselines, such as [11] or [12] in the reference. If these methods are not applicable to the experiments considered in the paper because the matching happens in the input space, you can try to compare the proposed method on the usual task that [11] or [12] experimented with. This would further highlight the benefit of doing matching in the latent space (even when the matching in the input space is feasible).

I wonder the proposed idea can also be adapted to unsupervised problems such as set autoencoders or flow-based models for sets. It is often hard to define proper likelihood functions for generative models for sets, so the proposed idea of matching in latent space can be a nice way to define likelihood functions.

**Time Spent Reviewing:**

6 hours

---

> ### Author Response · Authors · 2021-08-10
> **Response to the review**
>
> Thank you for your review. Below is our response to your comments.
>
> **Q1: It would be better though to include more recent baselines, such as [11] or [12] in the reference. If these methods are not applicable to the experiments considered in the paper because the matching happens in the input space, you can try to compare the proposed method on the usual task that [11] or [12] experimented with. This would further highlight the benefit of doing matching in the latent space (even when the matching in the input space is feasible).**
>
> We agree that comparison to a non-Hungarian set prediction method like [11] would be interesting. As for DSPN [12], we would like to point out that DSPN was outperformed by TSPN [1] which was included as a baseline already. For example, in the object detection task, we used DETR-based models which are Transformers that conform with the TSPN proposal. Hence, we did not include DSPN as a baseline.
>
> ---
> **Q2: I wonder the proposed idea can also be adapted to unsupervised problems such as set autoencoders or flow-based models for sets. It is often hard to define proper likelihood functions for generative models for sets, so the proposed idea of matching in latent space can be a nice way to define likelihood functions.**
>
> We are not familiar with flow-based models for sets, but LSP is certainly applicable to set autoencoders. Regarding the likelihoods, we noticed that set likelihood functions can be approximated by Hungarian assignment for non-sequence set elements [9]. We agree that LSP may allow one to define a likelihood function on the latent space for more complex set elements, such as sequences.
>
> [9] Hamid Rezatofighi, Roman Kaskman, Farbod T Motlagh, et al., “Learn to predict sets using Feed-Forward neural networks,” Jan. 2020.

---

### Official Review · Reviewer_vNZv · 2021-07-19

**Rating:** 6
**Confidence:** 4

**Summary:**

A major component of the set prediction task has to do with the computation of a matching loss between the predicted set and the original set.  In this paper,  the authors propose to instead compute this matching loss in a latent space. Computing the matching loss in latent space introduces some computational problems such as computational stability. Additionally, computing the matching loss in a latent space allows teacher forcing in the set prediction task. To tackle this, the authors outline sufficient conditions for necessary for computational stability as well as an efficient algorithm that improves model convergence in the    set prediction task. The authors perform experiments on two set prediction tasks including image captioning and object detection.

**Limitations And Societal Impact:**

The limitations and societal impacts of this work have been adequately described by the authors.

**Main Review:**

The presented model takes a new look at the matching loss used in many set prediction problems. By matching the set elements with predictions in a latent
embedding space, two main advantages are gained:
  - Firstly, it is no longer necessary to hand-craft distance metrics. A simple choice such as the Euclidean distance suffices.
  - Secondly, this matching loss can be efficiently used for teacher forcing with O(N) complexity compared to the O(N^2) requirement of current algorithms.
These contributions ensure that the tedious task of finding the right balance between multiple loss functions is alleviated. The introduction of
matching in the latent space ensures that even a simple choice such as the Euclidean distance metric can provide better model convergence with
theoretical guarantees albeit for no non-neural network based models.

While the presented method is theoretically sound, i have a few questions regarding its practicality.
 - How stable is the training process compared to the original matching loss not performed in latent space. My curiosity stems from the fact that in
   addition to the task loss, 4 new loss terms have now effectively been introduced. The first two are the asymmetric latent loss between the latent
   and guiding vectors. The remaining two, gradient cloning (GC) and gradient cloning with rejection (GCR) are terms added to ensure stability during
   training and a choice has to be made between GC and GCR. How long does this model have to be trained when compared with the vanilla model? Is the
   presented model more noisy during training? Perhaps showing convergence plots will be instructive? If so, how easy or difficult is it to tuned the strengths of these additional loss terms. In the
  appendix, it is stated that no hyperparameter search was performed but how were the loss balancing terms chosen?

Baselines:
  There are may recent works on set prediction that all use the traditional matching loss such as Object-Centric Learning with Slot Attention by
Locatello et. al. It will be instructive to see how the proposed loss
improves or degrade the performance on these models. As it stands, the baselines are just variations of the same method.

In Figure 1, the loss terms GC and GCR are not defined. Indeed the figure is referenced before these loss terms are defined at the end of Section 3.2.
It would be better to to make these loss terms clear in the caption to avoid any confusion.
Figures 2 and 3 are hard to interpret. It will be better to expand on the caption to make it clearer whats is being communicated.

**Time Spent Reviewing:**

6

---

> ### Author Response · Authors · 2021-08-10
> **Response to the review (Part 1 of 3)**
>
> Thank you for your review. Below is our response to your comments.
>
> **Q1: “in addition to the task loss, 4 new loss terms have now effectively been introduced. The first two are the asymmetric latent loss between the latent and guiding vectors. The remaining two, gradient cloning (GC) and gradient cloning with rejection (GCR) are terms added to ensure stability during training”**
>
> We would like to clarify that GC and GCR are not loss terms. They are modifications to the backpropagation paths to allow gradients to pass through the non-differentiable Hungarian assignment step in such a way that the convergence of the main task loss is ensured based on assumptions.
>
> ---
> **Q2: “a choice has to be made between GC and GCR.”**
>
> Our suggestion in the paper is to use GCR (d = $10^{-3}$) which was found to work well in all cases. Unless one is certain about the design of the encoder, GCR is always a preferable choice. The parameter “d” may need to be tuned as a hyperparameter.

---

> ### Author Response · Authors · 2021-08-10
> **Response to the review (Part 2 of 3)**
>
> **Q3: “How stable is the training process compared to the original matching loss not performed in latent space.”**
>
> The training process of LSP with GC or GCR is quite stable even when the encoder is suboptimal. The model may arrive at suboptimal solutions when the encoder is poor but no instability or divergence was observed in practice.
> For comparison with traditional set prediction approaches, please see response **Q4** below for more details.
>
> ---
> **Q4: “How long does this model have to be trained when compared with the vanilla model? Is the presented model more noisy during training? Perhaps showing convergence plots will be instructive?”**
>
> We agree that including convergence plots will be helpful. Based on our experiments, we did not observe any instabilities during the training of LSP given a reasonable encoder. For suboptimal encoders, the training was still stable but resulted in suboptimal performances.
>
> In the CLEVR task, although it should be noted that it is not a perfect comparison because the baselines are not set prediction methods, [(click to see image)](https://i.ibb.co/r4Sk29T/lsp-clevr.png) are the validation performance curves for LSP and the two baselines.
>
> We observed that Ordered Set and Concat converged faster than LSP but to worse solutions. Since these methods did not converge to solutions of the same quality, it was unfair to compare the convergence time directly. However, at any point in time, LSP was either on par or better with the other methods.
>
> In the object detection task, the best performing batch size was 32 for DETR and 8 for LSP. The training progresses are shown in [(click to see image, same batch size 8)](https://i.ibb.co/TtbG8Z7/lsp-detr-bs8.png) and [(click to see image, best batch sizes)](https://i.ibb.co/Y8nSfWr/lsp-detr-best-bs.png).
>
> No instability nor slowness in convergence was observed for LSP and DETR. It should be noted that AP (large) score is not shown because there was no large object. DETR (with batch size of 32) and LSP (with batch size of 8) were trained for the same number of iterations to not give LSP an advantage due to the smaller batch size. We will include the training progression curves in the appendix.

---

> ### Author Response · Authors · 2021-08-10
> **Response to the review (Part 3 of 3; final)**
>
> **Q5: “how easy or difficult is it to tuned the strengths of these additional loss terms. In the appendix, it is stated that no hyperparameter search was performed but how were the loss balancing terms chosen?”**
>
> There are two hyperparameters $\beta$ (Equation 2) and $d$ (for GCR):
>
> 1. $\beta$ was set to 0.1 without tuning based on a preliminary experiment on the CLEVR dataset. We found that $\beta = 0$ yielded slightly worse performance than $\beta = 0.1$ and that increasing beta further provided no significant differences. However, too large betas can slow down the training of the main loss function because the latent loss can compete with the main loss. For this reason, we kept the beta small. Setting $\beta = 0.1$ also worked well across experiments.
>
> 2. For $d$ in GCR, various values (on a log scale) were explored in each experiment. We found that the optimal value for $d$ can be different from task to task but setting $d = 10^{-3}$ worked reasonably well across tasks. However, we recommend that $d$ should be tuned on a log scale.
> In summary, we do not think that the choice of beta is of a particular concern while the choice of d can be more tricky. However, the most difficult component is not the loss function but the encoder (please see discussion in Section 6).
>
> ---
> **Q6: “There are may recent works on set prediction that all use the traditional matching loss such as Object-Centric Learning with Slot Attention by Locatello et. al. It will be instructive to see how the proposed loss improves or degrade the performance on these models. As it stands, the baselines are just variations of the same method.”**
>
> We completely agree that the Slot attention by Locatello et al. is an interesting work and that it would be useful for the readers to see how well LSP pairs with various model architectures. However, because LSP can be applied on top of any architecture, we focused on showing how LSP improves a common set prediction pipeline in general rather than showing how LSP works with a particular architecture.

---

> ### Comment · Reviewer_vNZv · 2021-08-24
> **Post Rebuttal Comment.**
>
> On the whole I am satisfied with the answers the authors have provided to my questions. Having read the questions and response to the other Reviewers, I agree that evaluation (specifically the choices of experiments as well as the choice of methods compared) can be improved. However I still like the contributions made by the authors in their novel approach to the set prediction problem and hence elect to maintain my score.

---

### Official Review · Reviewer_kmBN · 2021-08-01

**Rating:** 6
**Confidence:** 3

**Summary:**

This paper proposes LSP, an approach to set-to-set prediction that doesn’t require handcrafting a distance metric. This distance metric is often needed to perform bijective matching (e.g., the Hungarian algorithm) between the predicted set and the ground-truth set. The key idea is to perform the matching in the latent space instead, which means that encoding each element of the ground-truth set to this latent space has to be performed. Further, the authors adds two “tricks” to help with convergence: asymmetric latent loss (Eq. (2) with hyperparameter beta and gamma)) and gradient cloning with and without rejection (GC & GCR, with hyperparamter d to control the degree of rejection.)

The paper also provides the convergence analysis and experimental validation on 1 synthetic and 3 real datasets.

**Limitations And Societal Impact:**

Yes. I consider Section 6 is to be well thought-out and well-written.

**Main Review:**

### Motivation & Soundness & Significance of LSP

Overall, the authors do a great job discussing drawbacks of existing techniques and motivating and describing the proposed approach (L31-50). The method is sound, and both the asymmetric latent loss and the gradient cloning are nice additions that show that making the method “work” is not trivial.

But as I am not an expert on set prediction, it is difficult for me to judge the significance of the proposed change from the original feature space to the latent space. Things that will probably help to clarify this:
LSP1: One question I have is whether the loss in Eq. (1) would still need to be “designed”? Furthermore, the paper doesn’t provide much discussion or experimental validation to verify if set-to-set prediction is sensitive to the choice of distance metrics (e.g., how robust is [2] if you combine different l1 and CE loss in other ways? (L38-39).
LSP2: L36: I am not sure I understand the statement about vanilla GAN’s loss (“in terms of progress or distance”).
LSP3: L43-50: Are the authors saying that no existing work has done set prediction on sequence domains?

### Convergence analysis

I have not checked the correctness of this part carefully. A lot of details are in the supplement. However, it is quite clear that there is a gap between theory and experiments; for example, betas are different (L136-137, 167); the assumptions of the proof cannot be satisfied with neural networks (L309-311). This means that it is unclear how much of this theory explains the results, and the convergence analysis contribution wouldn’t be counted as a major one.

### Experiments

Two things I like about the experiments are the synthetic experiments and the fact that the authors look into diverse applications. My concerns are
E1: The experiments have little correlation with the authors’ claims in L59-62. What if you design the distance metrics and do not work with the latent space? Is O(N^2) -> O(N) improvement observed in practice?
E2: Similar to E1, it would be nice to have a small set of  experiments on asymmetric latent loss in addition to showing that GCR is better than GC. Further, it would be nice to have LSP without GC at all.
E3: Baselines are quite weak. Both “Concat” and “Ordered set” (L241-244) are not set prediction methods. Even DETR [2] is a modified version of LSP (L290-291).
E4: The CLEVR and MIMIC-CXR, though interesting, are proposed without much justification. Could the authors make it clear why looking at these instead of the standard tasks in the literature (if any)?
E5: Why is there no GCR in Fig. 4(a)?

### Clarity

The paper is overall clearly written. Figures are useful. The part that I really appreciate is Section 6, which clearly discusses limitations.

### Overall assessment
As I am trying to gain more clarity on motivation/significance of LSP and the details of the experiments, I think this paper is currently below the acceptance bar but I’m willing to be convinced otherwise.

### Post-rebuttal
The authors address many of my concerns, with additional experiments and clarifications about baselines. I still have concern about the fact that the authors haven't really showed/have evidence that designing distance metrics in the output space is really a problem. Please see the authors' response to my Q2. But, putting everything together, these new results push the paper above the acceptance bar, and I adjusted my score to 6. I would encourage the authors to motivate the paper even further.


**Time Spent Reviewing:**

2.25

---

> ### Author Response · Authors · 2021-08-10
> **Response to the review (Part 1 of 5)**
>
> Thank you for your review. Below is our response to your comments.
>
> **Q1: “LSP, an approach to set-to-set prediction”**
>
> LSP works with any input. We made almost no assumption on the model architecture. As long as the set elements are represented as latent vectors, LSP should be applicable.
>
> **LSP1: One question I have is whether the loss in Eq. (1) would still need to be “designed”?**
>
> The distance in Eq. (1) does not need to be designed. By operating directly on the latent space R^c in which squared Euclidean distance is naturally defined, LSP eliminates the need to design task-specific distance in Eq. (1). This is in contrast with other approaches that operate on task-specific output space, such as DETR [2] which combines L1 bounding box distance with “negative class probability” instead of using a more natural “negative log class probability” (i.e., cross-entropy loss).
>
> [2] Nicolas Carion, FranciscoMassa, Gabriel Synnaeve, et al., “End-to-End object detection with transformers,” May 2020.
>
> **Q2: “the paper doesn’t provide much discussion or experimental validation to verify if set-to-set prediction is sensitive to the choice of distance metrics (e.g., how robust is [2] if you combine different l1 and CE loss in other ways?”**
>
> We apologize for the missing information. We have preliminarily evaluated the variation in DETR’s performance by varying the weights of the distance function, that is  L1 bounding box distance and negative class confidence, and found that the performance changes by only around 1%. Overall, the result that LSP outperforms DETR holds especially on bounding box prediction (Table 4, AP (S) and AP75). This suggests that the manually designed DETR distance function is suboptimal. These details will be included in the final version of the manuscript.
>
> **LSP2: L36: I am not sure I understand the statement about vanilla GAN’s loss (“in terms of progress or distance”)**
>
> We are sorry for the confusion. GAN loss was mentioned as an example of a loss function that is inappropriate as a distance metric. While the scalar value of a good distance metric should reflect how “far” or how “close” two objects are, the vanilla GAN loss proposed by Goodfellow et al. is almost always constant during a normal training session. Its value does not indicate how close to the goal the prediction is. Although this kind of loss function is appropriate for training, it is not appropriate as a distance metric.

---

> > ### Comment · Reviewer_kmBN · 2021-08-29
> > **Clarification questions on your response to Q2**
> >
> > Thank you for very detailed responses. I wanted to clarify your response to Q2.
> > - Does the fact that the performance changes by only 1% a negative result (i.e., 1% = not sensitive to the weights of the distance function)?
> > - Which direction is this performance change?
> > - Instead of simply varying the weights of the distance function, do you think there can be other choices for distance metrics that are more optimal (for example, not l1 for bbox distance)?
> > - I'm not following how Table 4 is related to this and how it suggests "the manually designed DETR distance function is suboptimal"; would the authors mind clarifying?

---

> > > ### Author Response · Authors · 2021-08-30
> > > **Clarification to Q2**
> > >
> > > **Does the fact that the performance changes by only 1% a negative result (i.e., 1% = not sensitive to the weights of the distance function)?
> > >  Which direction is this performance change?**
> > >
> > > We don't want to claim either way since it was preliminary, yet believe 1% to be insignificant. This does not mean it is "not sensitive to the weights of the distance function" though, it's only **with respect to** small range of values we tried.
> > >
> > > We observed **worse** performance by changing the weights.
> > >
> > > **Instead of simply varying the weights of the distance function, do you think there can be other choices for distance metrics that are more optimal (for example, not l1 for bbox distance)?**
> > >
> > > It is a possibility. Usually, a designer takes inspiration from the loss function to design a distance function. It is unclear how to come
> > >  up with a very distinct yet good distance function while keeping the loss function the same because a distance function can also *harm* the performance especially when it is very distinct from the loss function itself.
> > >
> > > **I'm not following how Table 4 is related to this and how it suggests "the manually designed DETR distance function is suboptimal"; would the authors mind clarifying?**
> > >
> > > Table 4 compares DETR vs. DETR + LSP. The differences between the two models are only the assignment mechanism (output space vs latent space). The model architectures and even loss functions are identical (LSP has an additional latent loss, but the task loss is the same). The better performance observed from LSP cannot be explained by models or loss functions but only by the assignment mechanism which is effectively the distance function.

---

> ### Author Response · Authors · 2021-08-10
> **Response to the review (Part 2 of 5)**
>
> **LSP3: L43-50: Are the authors saying that no existing work has done set prediction on sequence domains?**
>
> What we want to convey is that although set prediction on the sequence domains is possible but not very practical when the number of elements, N, is large. There are some exhaustive approaches, $O(N^2)$, in multi-speaker automatic speech recognition (ASR), but those problems usually involve small N ($\leq 3$). For ASR, it is also possible to design a surrogate distance function via a CTC loss. However, in general, we have not seen any practical set prediction on general sequence domains with large N.
>
> **Q3: “it is quite clear that there is a gap between theory and experiments; for example, betas are different”**
>
> We would like to point out that the key requirement for convergence is the exponential reduction in distance between encoded vectors $g$ and set latent vectors $s$ (Lemma D.1). Even though the case with $\beta > 0$ is not included, because larger beta values would further narrow the distance between encoded vectors $g$ and set latent vectors $s$, it is very likely that Lemma D.1 will hold and that the rest of the proof will follow.
>
> **Q4: “This means that it is unclear how much of this theory explains the results, and the convergence analysis contribution wouldn’t be counted as a major one.”**
>
> While the convergence proof does not cover the general cases, it can still provide insights into how to design algorithms and encoders that will likely perform well. First of all, the proof shows that convergence is achieved if the algorithm can force the distance between encoded vectors $g$ and set latent vectors $s$ to eventually get smaller (Lemma D.1). This shows that “no GC” is a bad algorithm since it cannot satisfy this requirement. The theory also gives intuitions that the stronger latent loss is favorable and a better encoder is the one that is able to track the set latent vectors $s$ better. It also hints at what would happen if we fail to keep g and s close, the main loss function will not reduce beyond a certain point dictated by the distance of $g$ and $s$. This led us to propose techniques such as Gradient Cloning which help the encoded vectors $g$ stay close to the corresponding set latent vectors $s$ through the training process. The theory also suggests that we do not need to keep $g$ and $s$ infinitesimally close, we can get away with “less switch” which has the same effect as keeping the $g$ and $s$ distance small. Coupled with the fact that there is less switch in larger spaces, we can increase the latent dimensions to get better results as well. All in all, you can see that the theory provides a lot of intuitions and insights to the practitioners, no less than its theoretical contributions.

---

> ### Author Response · Authors · 2021-08-10
> **Response to the review (Part 3 of 5)**
>
> **E1: The experiments have little correlation with the authors’ claims in L59-62. What if you design the distance metrics and do not work with the latent space?**
>
> It is always possible to manually design a distance metric as has been done in previous works. We also agree that sound distance metrics can be straightforwardly obtained in many cases. However, manually designed metrics come with tradeoffs. For example, although the weighted sum of negative class prediction confidence and bounding box distance proposed by DETR is a sound choice, it cannot distinguish between instances with good class predictions but poor bounding boxes and instances with poor class predictions but good bounding boxes. With LSP, there is no such limitation because everything was decided by the gradients of the main objective function. This argument is partly supported by our experiments on object detection where LSP made better bounding box predictions than DETR (Table 4, AP (S) and AP75).
>
> Regarding L61-62, we want to clarify that the $O(N^2)$ problem has nothing to do with the problem of designing a distance metric. Teacher forcing is in conflict with Hungarian assignment because teacher forcing requires knowing the assigned ground truth for each element, while this information remains unknown until the assignment, which requires teacher forcing itself, is performed. There are two typical solutions: 1) do the teacher forcing with respect to all possible ground truths - this is the $O(N^2)$ problem, or 2) design a surrogate distance metric that does not require generation so that the assignment can be performed regardless of the teacher forcing. The former approach is impractical due to heavy resource requirements while the latter approach is far from trivial, especially in the general domain. LSP provides a third option for solving this problem. We have experimented with this scenario in the image captioning experiments, although we have not included an $O(N^2)$ baseline due to limited computing resources.
>
> **Q5: “Is O(N^2) -> O(N) improvement observed in practice?”**
>
> We have not benchmarked an $O(N^2)$ approach because of our limited resources, but we strongly believe that this improvement is observable in practice because both the time and memory requirements of the GPU will be reduced by the factor of N. On a related note, we also would like to point out that although the Hungarian algorithm is $O(N^3)$, it is fast because it is solved directly on an $N \times N$ matrix without performing expensive operations like forward passes on a deep model.

---

> ### Author Response · Authors · 2021-08-10
> **Response to the review (Part 4 of 5)**
>
> **E2: Similar to E1, it would be nice to have a small set of experiments on asymmetric latent loss in addition to showing that GCR is better than GC.**
>
> We agree that this would be an improvement to the manuscript and will be added to the manuscript. For the moment, we have rerun the CLEVR experiments (3 new seeds) to compare GC at various values of β as follows (± represents one standard deviation).
>
> | Method | Precision | Recall | F1 |
> | --- | --- | --- | --- |
> | GC (β = 0) | 0.979 ± 0.01 | 0.970 ± 0.02 | 0.975 ± 0.01 |
> | GC (β = 0.1) | **0.989** ± 0.01 | **0.979** ± 0.01 | **0.984** ± 0.01 |
> | GC (β = 0.2) | 0.987 ± 0.01 | 0.976 ± 0.02 | 0.982 ± 0.01 |
> | GC (β = 0.5) |  0.980 ± 0.01 | 0.972 ± 0.01 | 0.976 ± 0.01 |
> | GC (β = 1) | 0.983 ± 0.01 | 0.966 ± 0.02 | 0.974 ± 0.02 |
>
> Note that the differences are not significant given the variances. We cannot conclude that β = 0.1 is the best. We believe that there are other more important factors to be considered for tuning, such as the design of the encoders.
>
> ---
> **Q6: Further, it would be nice to have LSP without GC at all.**
>
> We will add this comparison to the manuscript. Our experience was that without GC, the model may not converge on some synthetic datasets. For the moment, we have rerun the CLEVR experiments to compare “no GC” and “GC” (3 new seeds) [(click to see image)](https://i.ibb.co/v1M7xwx/lsp-gc-vs-nogc.png)  and found that the performance of  the model without GC is more variable between initializations compared to the model with GC. Given that the set cardinality is not large (< 10 sentences) and a large latent dimension (256) was used, there was some possibility that the model “no GC” may converge in practice, as suggested by the result from synthetic dataset experiment (Section 5.1).
>
> Quantitatively (± represents one standard deviation):
>
> | Method | Precision | Recall | F1 |
> | --- | --- | --- | --- |
> | GC (β = 0.1) | **0.989** ± 0.01 | **0.979** ± 0.01 | **0.984** ± 0.01 |
> | no GC (β = 0.1)  | 0.976 ± 0.01 | 0.900 ± 0.06 | 0.936 ± 0.04 |
>
> There are substantial differences between “GC” and “no GC”.
>
> ---
> **E5: Why is there no GCR in Fig. 4(a)?**
>
> GCR and GC both converged robustly in all of these cases. In fact, GCR has a stronger convergence than GC, and without neural nets, even GC is already guaranteed to converge. We will revise the caption to state that both GC and GCR converged in all of these settings.

---

> ### Author Response · Authors · 2021-08-10
> **Response to the review (Part 5 of 5; final)**
>
> **E3: Baselines are quite weak. Both “Concat” and “Ordered set” (L241-244) are not set prediction methods. Even DETR [2] is a modified version of LSP (L290-291).**
>
> We would like to point out that before LSP, set-of-sequence prediction with teacher forcing for image captioning requires an exhaustive generation $O(N^2)$ which is impractical in terms of both GPU time and memory. This is also evident from the fact that other methods on image captioning [R1, R2] and medical report generation [R3, R4] are all non-set methods. They are also essentially the same as the “Concat” baseline. RM+MCLN [24], a recent work in chest x-ray, is also included in our comparison. Hence, we believe that our choices of baselines are reasonable and representative of the current literature.
>
> DETR [2], which is comparable to TSPN [1], was selected because it represents a strong baseline in object detection that uses set prediction. We deliberately made minimal modifications to DETR, by replacing just the part that does the assignment in the output space with LSP, in order to highlight the impact of performing assignment in the latent space. The results showed that LSP provides performance improvement even on an already strong baseline. We want to note that DSPN [12] was not a strong baseline (according to [1]) and was not widely used in the community, hence we did not include it here.
>
> **References:**
>
> [R1] Lu, Jiasen, Caiming Xiong, Devi Parikh, and Richard Socher. 2017. “Knowing When to Look: Adaptive Attention via a Visual Sentinel for Image Captioning.” In Proceedings of the IEEE Conference on Computer Vision and Pattern Recognition, 375–83.
>
> [R2] Huang, Lun, Wenmin Wang, Jie Chen, and Xiao-Yong Wei. 2019. “Attention on Attention for Image Captioning.” arXiv [cs.CV]. arXiv. http://arxiv.org/abs/1908.06954.
>
> [R3] Liu, Guanxiong, Tzu-Ming Harry Hsu, Matthew McDermott, Willie Boag, Wei-Hung Weng, Peter Szolovits, and Marzyeh Ghassemi. 2019. “Clinically Accurate Chest X-Ray Report Generation.” In Proceedings of the 4th Machine Learning for Healthcare Conference, PMLR, edited by Finale Doshi-Velez, Jim Fackler, Ken Jung, David Kale, Rajesh Ranganath, Byron Wallace, and Jenna Wiens, 106:249–69. Proceedings of Machine Learning Research. Ann Arbor, Michigan: PMLR.
>
> ---
> **E4: The CLEVR and MIMIC-CXR, though interesting, are proposed without much justification. Could the authors make it clear why looking at these instead of the standard tasks in the literature (if any)?**
>
> We looked for set prediction tasks that would highlight the new capabilities gained through LSP. First, the sets to be predicted should not be too small (otherwise non-set prediction would suffice). Second, teacher forcing, which was not possible in prior set prediction approaches, should be applicable. Third, the input should be highly predictive of the output to minimize effects from confounding factors. Lastly, we preferred tasks with real-world applications.
>
> MIMIC-CXR (chest x-ray reported generation) was selected because a medical report can be thought of as a set of sentences each describing an observed abnormality. We prefer MIMIC-CXR over standard datasets like MS-COCO because each MS-COCO’s caption usually contains one to two sentences while medical reports are much longer. Furthermore, MIMIC-CXR is less subjective than MS-COCO because while all radiologists receive standardized training to spot the same abnormalities, general image captioning highly depends on the labellers. Most importantly, medical reports [R4, R5] and automatic chest x-ray assessments [R1, R2, R3] have seen great interest in recent years due to its applications in healthcare [R6].
> CLEVR was selected mainly because its ground truths are known without any ambiguity nor confounding factor. Hence, it is perfect for comparing LSP against other methods.
>
> **References:**
>
> [R1] Rajpurkar, Pranav, Jeremy Irvin, Kaylie Zhu, Brandon Yang, Hershel Mehta, Tony Duan, Daisy Ding, et al. 2017. “CheXNet: Radiologist-Level Pneumonia Detection on Chest X-Rays with Deep Learning.” arXiv [cs.CV]. arXiv. https://doi.org/1711.05225.
>
> [R2] Irvin, Jeremy, Pranav Rajpurkar, Michael Ko, Yifan Yu, Silviana Ciurea-Ilcus, Chris Chute, Henrik Marklund, et al. 2019. “CheXpert: A Large Chest Radiograph Dataset with Uncertainty Labels and Expert Comparison.” Proceedings of the AAAI Conference on Artificial Intelligence 33 (July): 590–97.
>
> [R3] Horng, Steven, Ruizhi Liao, Xin Wang, Sandeep Dalal, Polina Golland, and Seth J. Berkowitz. 2021. “Deep Learning to Quantify Pulmonary Edema in Chest Radiographs.” Radiology: Artificial Intelligence 3 (2): e190228.
>
> [R4] Chen, Zhihong, Yan Song, Tsung-Hui Chang, and Xiang Wan. 2020. “Generating Radiology Reports via Memory-Driven Transformer.” arXiv [cs.CL]. arXiv. https://www.aclweb.org/anthology/2020.emnlp-main.112.pdf.
>
> [R5] Liu, Guanxiong, Tzu-Ming Harry Hsu, Matthew McDermott, Willie Boag, Wei-Hung Weng, Peter Szolovits, and Marzyeh Ghassemi. 2019. “Clinically Accurate Chest X-Ray Report Generation.” In Proceedings of the 4th Machine Learning for Healthcare Conference, PMLR, edited by Finale Doshi-Velez, Jim Fackler, Ken Jung, David Kale, Rajesh Ranganath, Byron Wallace, and Jenna Wiens, 106:249–69. Proceedings of Machine Learning Research. Ann Arbor, Michigan: PMLR.
>
> [R6] Sim, Yongsik, Myung Jin Chung, Elmar Kotter, Sehyo Yune, Myeongchan Kim, Synho Do, Kyunghwa Han, et al. 2020. “Deep Convolutional Neural Network-Based Software Improves Radiologist Detection of Malignant Lung Nodules on Chest Radiographs.” Radiology 294 (1): 199–209.

---

> > ### Comment · Reviewer_kmBN · 2021-08-29
> > **Your response to E3**
> >
> > "We deliberately made minimal modifications to DETR, by replacing just the part that does the assignment in the output space with LSP, in order to highlight the impact of performing assignment in the latent space."
> > - To make sure I understand this, is DETR [2] performing the assignment on the output space? In my original review, I understood L290-291 as DETR [2] in Table 4 = a modified DETR baseline with LSP.

---

> > > ### Author Response · Authors · 2021-08-30
> > > **Table 4 compares original DETR and DETR + LSP**
> > >
> > > We want to clarify that DETR [2] performs the Hungarian assignment on the output space. We used the **original** DETR as our baseline.
> > > We compare this baseline against LSP by **making minimal changes** to DETR (making assignment in the latent space).
> > > Both models have identical image encoder and transformer architecture overall.

---

> > > > ### Comment · Reviewer_kmBN · 2021-08-30
> > > > **Thanks**
> > > >
> > > > Thanks. I totally misread L290-291 before. That's why Table 4 didn't make sense to me. More clear to me now.

---

> ### Comment · Area_Chair_c68Q · 2021-08-22
> **Please provided feedback to the authors' rebuttal**
>
> Dear Reviewer kmBN,
>
> The authors have provided very detailed responses to your comments with additional experimental results. Could you please go over them and see whether they change your initial opinion on the paper? The decision deadline is quickly approaching and we need to reach a consensus soon.
>
> Thanks,
> Area Chair

---

### Decision · Program_Chairs · 2021-09-27

**Decision:**

Accept (Poster)

**Comment:**

This paper proposes a novel method for set prediction tasks, whose goal is to predict multiple elements without consideration of their orderings. A limitation of the existing set prediction methods is that they need to solve for the matching problem between the predicted and the ground-truth set under a certain distance metric, but the choice of distance metric is critical for the convergence property of the matching algorithm. To deal with this issue, the authors propose to learn a latent space and perform set prediction tasks in this space, which they refer to as Latent Set Prediction (LSP). LSP is beneficial over existing methods as it allows us to use simple Euclidean distance, which eliminates the need of selecting a specific hand-crafted distance metric, and enables efficient matching with teacher-forcing. However, a naive LSP may not converge well due to the instability of the matchings across the set elements, and the authors propose techniques to allow stable pairing of the elements across two sets, further showing its convergence guarantee. The authors validate LSP on semantic scene description, multi-modal report generation, and the object detection task, and the results show the effectiveness of the proposed LSP over relevant baselines.

The paper received split initial reviews with three leaning negative and two positive. However, despite the negative scores, most reviewers found the paper well-written, the tackled problem important, and the discussion on the limitations of the existing set prediction techniques insightful. Also, they considered the proposed set matching technique to be sound and nontrivial, and the provided theoretical convergence guarantee to be valuable.

However, the reviewers had the following common concerns:
There exists an obvious gap between theory and practice, which renders the convergence property of the proposed method unclear in real-world scenarios, but there is no empirical analysis of convergence.
The baselines used for the experiments are weak, and the proposed method is not validated against some highly relevant baselines for set prediction (e.g. TSPN and DSPN).
There is no ablation study of the proposed losses and techniques, which are essential in verifying their effectiveness.
The experimental validation part of the paper is not well organized, and comes with different baselines and evaluation protocols.

During the discussion period, the authors dealt away most of the concerns by providing experimental results with relevant baselines, providing the results of the ablation studies, and by providing the learning curves. The reviewers revised their reviews and increased their scores as they found the new results and discussions satisfactory, and reached a consensus to accept the paper.

I also agree with the reviewers that the paper is proposing a highly original idea and sound methods to solve a well-motivated problem, and that the theoretical analysis is nice. The only weak part was experimental validation, but this has been satisfactorily addressed in the author responses, and I believe that the paper will be in a very good shape after incorporating all the discussions and results into the revised version of the paper.